# Parvalbumin-expressing basal forebrain neurons mediate learning from negative experience

Panna Hegedüs[1,2,4], Bálint Király [1,4], Dániel Schlingloff [1,4], Victoria Lyakhova [1,2], Anna Velencei[1], Írisz Szabó[1], Márton I. Mayer [3], Zsofia Zelenak[1], Gábor Nyiri [3] & Balázs Hangya [1] ✉

Parvalbumin (PV)-expressing GABAergic neurons of the basal forebrain (BFPVNs) were proposed to serve as a rapid and transient arousal system, yet their exact role in awake behaviors remains unclear. We performed bulk calcium measurements and electrophysiology with optogenetic tagging from the horizontal limb of the diagonal band of Broca (HDB) while male mice were performing an associative learning task. BFPVNs responded with a distinctive, phasic activation to punishment, but showed slower and delayed responses to reward and outcome-predicting stimuli. Optogenetic inhibition during punishment impaired the formation of cue-outcome associations, suggesting a causal role of BFPVNs in associative learning. BFPVNs received strong inputs from the hypothalamus, the septal complex and the median raphe region, while they synapsed on diverse cell types in key limbic structures, where they broadcasted information about aversive stimuli. We propose that the arousing effect of BFPVNs is recruited by aversive stimuli to serve crucial associative learning functions.

Basal forebrain (BF) nuclei in the ventral part of the forebrain are defined by the presence of cholinergic projection neurons (BFCNs)[1–3]. However, there are approximately five times more GABAergic than cholinergic neurons in the BF[4–6], of which the parvalbumin-expressing population was shown to project to several cortical and subcortical target areas[7–9].

These BFPVNs were shown to be important regulators of cortical gamma oscillations[10,11] and to promote wakefulness and arousal through their local[12] and distant projections[13]. However, a number of observations point to the direction that BFPVNs might also play a role in awake behaviors, and associative learning in particular. First, similarly to BFCNs, degeneration of BFPVNs was also linked to cognitive decline in Alzheimer's disease[14] and during physiological aging[15–17]. Second, non-selective lesions of the BF revealed a more substantial

attentional and learning deficit in rodents compared to selective cholinergic lesions[18–21] and a study of selective BF GABAergic ablation revealed a deficit in eyeblink conditioning[22]. Third, a line of studies demonstrated that non-cholinergic BF neurons responded phasically to behaviorally salient stimuli irrespective of their valence[23], where stronger responses[24] or higher pre-stimulus activity[25] predicted faster and more precise decision speed during operant conditioning. These studies raise the possibility that BFPVNs might also play a role in associative learning.

To address this question, we performed in vivo electrophysiological recordings with optogenetic tagging and bulk calcium measurements by fiber photometry of PV-expressing neurons of the rostral HDB nucleus of the BF (referred to as BFPVNs hereinafter), while mice were performing a probabilistic Pavlovian conditioning

[1]Lendület Laboratory of Systems Neuroscience, HUN-REN Institute of Experimental Medicine, H-1083 Budapest, Hungary. [2]János Szentágothai Doctoral School of Neurosciences, Semmelweis University, H-1085 Budapest, Hungary. [3]Laboratory of Cerebral Cortex Research, HUN-REN Institute of Experimental Medicine, H-1083 Budapest, Hungary. [4]These authors contributed equally: Panna Hegedüs, Bálint Király, Dániel Schlingloff. ✉e-mail: hangya.balazs@koki.hun-ren.hu

task[26]. We found that BFPVNs showed rapidly rising, brief (i.e., phasic) responses to punishment, whereas rewards, as well as sensory stimuli that predicted reward or punishment, elicited comparably smaller and delayed responses. These observations revealed that BFPVN activity correlated with behavioral reinforcement, and punishment in particular, suggesting that these neurons signaled negative behavioral outcome to other parts of the brain. In accordance, optogenetic inhibition of BFPVNs during punishments in a Pavlovian conditioning task impaired the ability of mice to form stimulus-outcome associations, demonstrating that HDB BFPVNs encode outcome-related information crucial for associative learning.

We performed cell type-specific anterograde and mono-transsynaptic retrograde tracing experiments to reveal long-range afferent and efferent connections of HDB BFPVNs. We demonstrated that they received dense inputs from the lateral hypothalamus, the septal complex and the median raphe nucleus, and innervated important nodes of the limbic system crucial for contextual learning, including the medial septum, the hippocampal formation and the retrosplenial cortex[27–32]. They synapsed on a range of cell types including cholinergic and PV-expressing neurons in the BF, as well as interneuron types of the neocortex and hippocampus, as revealed by light- and electron microscopy and in vitro acute slice recordings. Finally, by performing bulk calcium measurements of HDB BFPVN fibers in target areas, we demonstrated that these neurons broadcast outcome-related information to multiple downstream areas. These data suggest that BFPVNs of the HDB mediate learning from negative reinforcement.

## Results

### Identified BFPVNs of the HDB show phasic responses to punishment

We trained PV-Cre mice ($n = 4$) on an auditory probabilistic Pavlovian conditioning task (Hegedüs et al., 2021) to examine how HDB BFPVNs respond to cues predicting reward or punishment with different probabilities and to the reinforcement itself. Mice were head restrained and two tones of different pitches were presented in a randomized order (Fig. 1a); Cue 1 predicted reward with higher probability (water drop; 80% reward, 10% punishment, 10% omission), whereas Cue 2 predicted punishment with higher probability (air puff; 65% punishment, 25% reward, 10% omission). This experimental design enabled us to examine how BFPVNs respond to surprising and expected reward or punishment (unconditioned stimuli, US) as well as their predicting cues (conditioned stimuli, CS). Mice learned these task contingencies, demonstrated by significantly higher anticipatory lick rate in response to Cue 1 that predicted likely reward (Fig. 1b–e).

To examine how individual BFPVNs responded to conditioned and unconditioned stimuli, we injected AAV2.5-EF1a-DiO-hChR2(H134R)-eYFP.WPRE.hGh into the HDB region of PV-Cre mice and performed extracellular multiple single-cell recordings combined with optogenetic tagging of HDB BFPVNs ($n = 36$) by implanting a head-mounted micro-drive housing moveable tetrode electrodes and an optical fiber (Fig. 1f–j, Fig. S1; specific expression verified in ref. 33, Fig. S15). Electrode placement in the HDB was verified by histological reconstruction at the end of the experiment (Fig. 1f, g). We found that BFPVNs responded with a gradual, ramp-like activation to cue onset ($n = 12/36$, 33%; Fig. 2a, b) irrespective of the predictive value of the cues (Fig. S2) and with a small but significant activation to the delivery of water reward ($n = 14/36$, 39%; Fig. 2c, d). In contrast, they responded to punishment with fast, phasic activation, homogeneously detectable in most ($n = 27/36$, 75%) identified BFPVNs (Fig. 2e, f, Table S4). BFPVN punishment responses showed adaptation, as they decreased in magnitude as a function of repeated air puff presentations; however, they remained present in late trials of the training sessions (Fig. 2g–i). BFPVN responses to reward and punishment were not modulated by

surprise (Fig. S2), unlike what we found in the case of reward-elicited responses of BFCNs in the same task[34,35].

An unbiased clustering approach applied on the response patterns of $n = 685$ HDB neurons revealed two clusters of neurons ($n = 134$ and 193) exhibiting strong punishment and smaller reward responses similar to the identified HDB BFPVNs, which indeed largely fell in these clusters (29/36, 80%; Fig. S3). A distinct group ($n = 97/685$, 14%) showed suppression of firing after reinforcement, while one cluster ($n = 48/685$, 7%) corresponded to putative cholinergic neurons based on their fast responses to air puffs[25,36] and slower responses to rewards and reward-predicting stimuli[35], completed with a mostly non-responsive cluster of cells ($n = 213/685$, 31%).

We also contrasted HDB BFPVNs with another prominent GABAergic population of the basal forebrain, the somatostatin-expressing neurons (BFSOMNs), by performing bulk calcium measurements from the HDB of task-performing SOM-Cre mice injected with AAV2/9.CAG.Flex.GCaMP6s.WPRE.SV40 (Fig. S4). BFSOMNs showed distinct responses compared to BFPVNs, with comparable responses to reward and punishment modulated by outcome expectations, and a phasic response to reward-predicting stimuli.

We examined the electrophysiological properties of punishment responsive BFPVNs and found that half of these neurons were burst firing. Both burst spikes and single spikes were time-locked to punishment presentation, but bursting BFPVNs were more punishment-activated ($n = 17/18$, 94%) than non-bursters ($n = 10/18$, 56%; Fig. S5).

### HDB BFPVNs respond to aversive stimuli across sensory modalities

We asked whether HDB BFPVNs respond to aversive stimuli in multiple sensory modalities. Therefore, we tested their responses to air puffs characterized by tactile sensation and loud noise[26] as well as foot shock and predator (fox) odor by performing bulk calcium measurements from the HDB of mice expressing GCaMP6s in BFPVNs. Since these stimuli are innately aversive to mice, we expected that they evoke neural responses already at first encounters, similar to what we found previously in BFCNs[25].

BFPVNs responded to air puffs as early as during training sessions when punishments were introduced, and even when only the first punished trials were considered (Fig. 3a). Like air puffs, foot shocks and the presence of fox odor also potently activated BFPVNs, suggesting that they convey information that generalizes over different aversive events (Fig. 3b, c). These responses also occurred already at the first encounter. Activation by air puffs, electrical shocks and predator odor indicate that BFPVN responses to aversive stimuli are multimodal, and not solely driven by a single sensory modality.

### Optogenetic inhibition of HDB BFPVNs during punishment impairs association learning

To assess whether the robust responses of BFPVNs to punishment represent an aversive signal that induces avoidance behavior, we tested whether optogenetic activation of BFPVNs elicited conditioned place aversion. We injected either AAV2.5-EF1a-DiO-hCHR(H134R)-eYFP.WPRE.hGh ($n = 8$) or AAV2.5-EF1a-DiO-eYFP.WPRE.hGh ($n = 10$ used as controls) viral vectors into the HDB of PV-Cre mice and performed a conditioned place aversion test. Mice were placed in an arena, the two sides of which were separated by a small open gate and differentiated by dotty and striped sidewall patterns. After habituation, BFPVNs were optogenetically stimulated in one side of the arena and different behavioral parameters such as total distance traveled, speed, time spent on either side of the arena, grooming, digging, freezing, and rearing were recorded and analyzed. We found that stimulation of HDB BFPVNs did not elicit aversive (or appetitive) behavior, as channelrhodopsin-expressing mice did not avoid the stimulated side, nor did they express any other behavioral differences

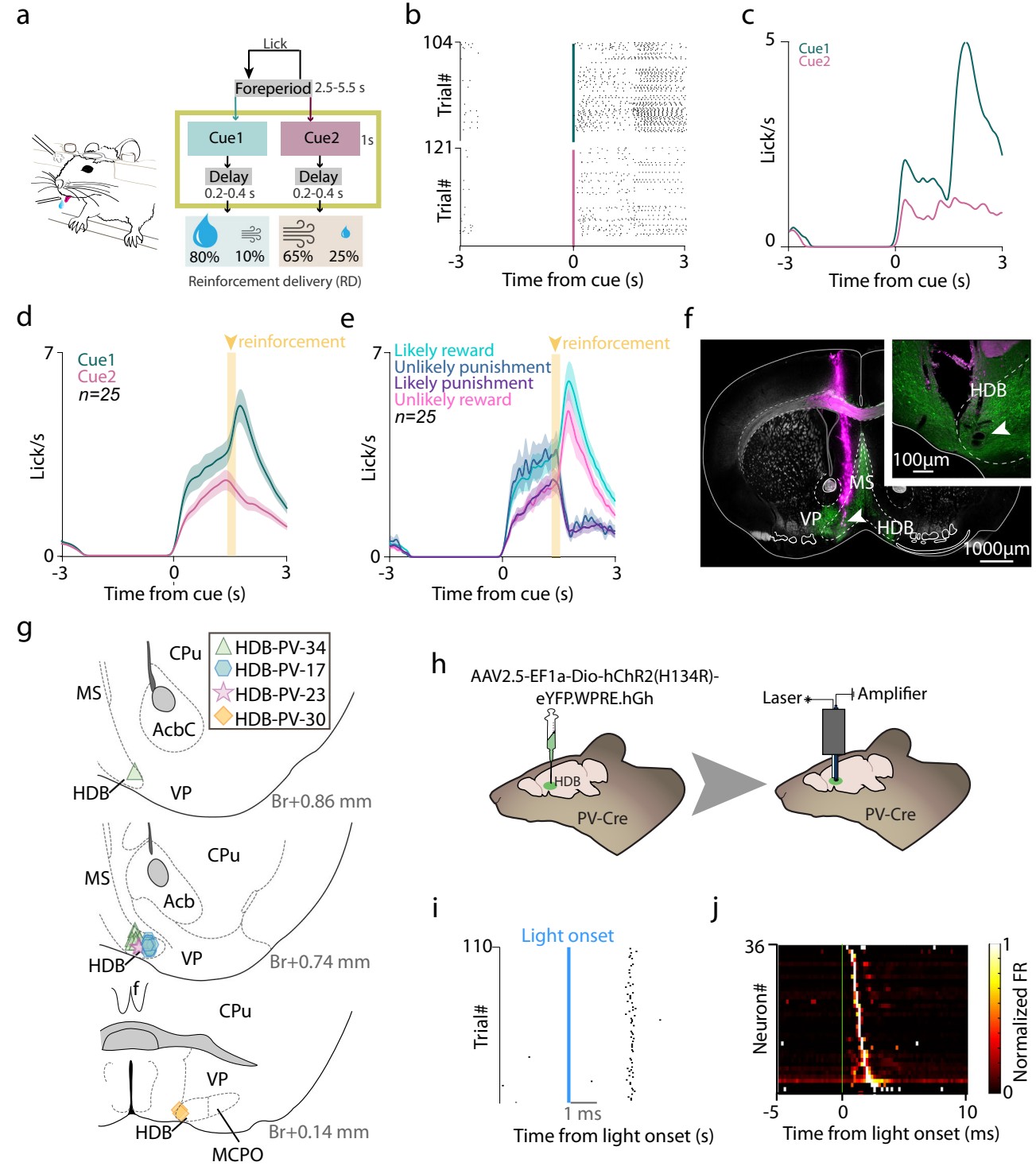

compared to control animals, and they did not show behavioral signs of stress (Fig. S6).

Next, to causally test the behavioral function of the punishment-elicited HDB BFPVN response in learning, we tested whether the optogenetic suppression of BFPVNs during the time of air puff delivery affected the learning ability of mice. We injected the HDB of PV-Cre mice bilaterally with AAV2.9-CAG-Flex-ArchT-GFP ($n = 6$, ArchT group) or AAV2.9-CAG-Flex -GFP ($n = 8$, control group; Fig. 4a). A one-second-long continuous laser pulse was applied to HDB BFPVNs through bilaterally implanted optic fibers starting at the same time as punishment presentation while mice were trained on the auditory Pavlovian conditioning task. Both ArchT-expressing and control mice developed

anticipatory licking in response to the conditioned stimuli. However, while control mice learned to suppress licking after Cue 2 which signaled likely punishment (similar to our initial cohort, see Fig. 1b–e), lick rates of ArchT-expressing mice remained high after both cues (Fig. S7) and thus they did not show differential anticipatory licking after the CS (Fig. 4b, c). Receiver operating characteristic analysis (ROC; see Methods) revealed a significant difference between ArchT-expressing and control mice in a half-second window before reinforcement delivery (from about 0.6 to 1.1 s after cue onset, Fig. 4b, c, Fig. S7). Control mice not only responded with a higher anticipatory lick rate to Cue 1 predicting likely reward, but their responses were significantly faster compared to those after Cue 2 predicting likely punishment.

**Fig. 1 | Head-fixed probabilistic Pavlovian conditioning combined with tetrode recording and optogenetic tagging. a** Schematic diagram of the task protocol. After a variable foreperiod, in which the animals were not allowed to lick, they were presented with two cues, either predicting likely reward or likely punishment; reinforcement was preceded by a short, variable delay period. Created using Mathis, M. (2020), Classical Conditioning Mouse, Zenodo, https://doi.org/10.5281/zenodo.3925907, under Creative Commons 4.0 license (https://creativecommons.org/licenses/by/4.0/). The original image was not modified. **b** Raster plot of mouse licking times aligned to stimulus (cue) onset from an example session, showing higher anticipatory lick rate in response to the reward-predicting cue (Cue 1). **c** Lick rates (peri-event time histograms, PETH) aligned to stimulus onset from the same example session. **d** Average lick rates of all recording sessions in which BFPVNs were identified ($n = 25$ sessions; errorshade, SEM). **e** Average lick rates of the same $n = 25$ sessions as in (**d**), partitioned according to the four possible outcomes (errorshade, SEM). **f** Histological reconstruction of the tetrode track in the HDB.

Green, ChR2-eYFP; magenta, DiI. Top right, magnified image of the electrolytic lesion site (white arrowhead) from a different section, indicating the tip of the electrodes. MS, medial septum; VP, ventral pallidum. Repeated in $n = 4$ mice. **g** Locations of the recorded BFPVNs based on histological reconstruction. Different markers correspond to different mice. Numbers show antero-posterior positions relative to Bregma. **h** Schematic illustration of viral injections, extracellular tetrode recordings and optogenetic tagging in mice. Created using Kennedy, A. (2020), Mouse brain silhouette, Zenodo, https://doi.org/10.5281/zenodo.3925919, under Creative Commons 4.0 license (https://creativecommons.org/licenses/by/4.0/). The original image was modified by adding illustrations of a syringe and an electrophysiological drive as well as additional legends. **i**, Example raster plot of spike times aligned to photostimulation pulses of an optogenetically identified BFPVN. **j** Firing rate (color-coded PETH) of all optogenetically identified BFPVNs aligned to photostimulation onset.

---

This reaction time difference was also abolished by suppressing HDB BFPVNs in the ArchT group of mice (Fig. 4d, e). Thus, optogenetic suppression of HDB BFPVNs prevented mice from associating higher value with the cue that predicted likely reward.

## BFPVNs of the HDB receive inputs from aversion-coding nuclei and send projections to the limbic system

We used mono-transsynaptic rabies tracing to identify brain areas and cell types that provide synaptic input to HDB BFPVNs. We injected Cre-dependent helper virus and G protein-deleted EnvA-pseudotyped rabies virus into the HDB of PV-Cre mice ($n = 3$; Fig. 5). Long-range input cells were distributed throughout the whole brain, suggesting a high level of input convergence onto HDB BFPVNs (Table S1.). The hypothalamus provided the most inputs (lateral hypothalamus, 33.1%, fraction of total input neurons; preoptic area, 7.6%), followed by the lateral septum (13.8%). Neighboring areas of the basal forebrain, such as the medial septum (MS, 10.8%) and the vertical limb of the diagonal band of Broca (VDB, 6.4%) also contained substantial number of input cells, together with the nucleus accumbens (3.5%), while the median raphe region (MRR) provided the highest number of synaptic inputs from the brainstem (3.9%, Fig. 5e). In case of one mouse, we performed unilateral helper and rabies virus injection to the HDB. In this animal, the presynaptic neurons were predominantly ipsilateral (93.7%) to the starter population, and the pattern of anatomical connectivity was similar for ipsilateral and contralateral projections.

Because MRR has an important role in mediating negative experience, we further investigated its cell types that provided inputs to HDB BFPVNs. MRR vesicular glutamate transporter 2 (vGluT2) expressing cells that are key regulators of negative experience are known to target BFPVNs[37]. However, MRR has several other types of cells including serotonergic (5HT) neurons, vGluT3-expressing glutamatergic neurons, those that express both serotonin and vGluT3, and GABAergic neurons as well[38]. We found that most of the input cells of BFPVNs were vGluT3-expressing glutamatergic neurons (59%, $n = 2$ mice) and out of these cells only a small fraction was serotonergic as well (5-HT-immunopositive, 2% of all inputs from MRR, Fig. 5f, g). We found no input cells that were only 5HT-immunopositive. The rest (41%) of the input cells (that were immunonegative for both vGluT3 and 5-HT) included vGluT2 cells that are known to target BFPVNs and may also include some MRR GABAergic neurons. These suggest that MRR targets BFPVNs with both vGluT2- and vGluT3-expressing glutamatergic neurons, both of which play a key role in the brainstem control of negative experience[37,39].

Next, we examined the downstream targets of HDB BFPVNs by performing a series of anterograde tracing experiments. We injected AAV2.9-CAG-Flex-eGFP viral vectors bilaterally in the HDB of PV-Cre mice to visualize HDB BFPVN axons using fluorescent microscopy ($n = 3$; Fig. 6a). We found that the majority of the projecting axons targeted the medial septum−diagonal band of Broca area, the hippocampus and the retrosplenial cortex (RSC; Fig. 6a), which are important regions of the limbic system. Additionally, other limbic structures such as the medial mammillary nucleus, the supramammillary nucleus and the paratenial thalamus were also targeted by HDB BFPVNs. A smaller extent of projections was received by the prefrontal cortex (medial orbital and infralimbic cortices), the lateral septum and the MRR. By using correlated electron and light microscopy, we demonstrated that HDB BFPVNs established symmetrical synaptic contacts with PV-expressing and cholinergic neurons in the medial septum, with calretinin-expressing (CR+) and PV-expressing neurons in the hippocampal CA1 region and with PV-expressing neurons in the retrosplenial cortex (Fig. 6b–d).

## HDB BFPVNs broadcast negative reinforcement signals to multiple downstream targets

Finally, we tested whether HDB BFPVNs send signals upon aversive US in a target-specific manner or distribute similar information to multiple downstream areas. To address this, we expressed GCaMP6s in PV-Cre mice ($n = 12$) by injecting AAV2/9.CAG.Flex.GCaMP6s.WPRE.SV40 bilaterally in the HDB and recorded their bulk calcium signals both at the cell body level within the HDB and in axonal projections in their main target areas such as the MS, the hippocampal CA1 and the retrosplenial cortex (Fig. 7a). We recorded bulk calcium signals via fiber photometry while mice performed the Pavlovian conditioning task.

First, we confirmed that somatic calcium concentrations showed similar responses to air puff punishments to what we observed when single neuron spiking responses of BFPVNs were analyzed (Fig. 7b, c, Fig. S8). Next, we found that BFPVN projections showed phasic punishment responses in all three target areas investigated (Fig. 7d–i), arguing for a widespread broadcast-type function of HDB BFPVNs.

Next, to test the physiological impact of these BFPVN projections on target neurons, we performed channelrhodopsin-assisted circuit mapping experiments in the dorsal CA1, medial septum and retrosplenial cortex (Fig. 8a–c). These experiments confirmed all connections revealed by EM (PV+ cells in RSC, CA1 and MS, CR+ in the hippocampus and ChAT+ cells in the MS; Fig. 8d–f). Furthermore, we found that BFPVN inhibitory inputs to the RSC showed smaller amplitude compared to the MS and CA1 as well as longer latency to peak and slower rise time (Fig. 8g). Importantly, when we examined the short-term synaptic plasticity of BFPVN inputs, we found that inputs to the RSC neurons were characterized by short-term synaptic facilitation, while MS and CA1 neurons showed strong short-term synaptic depression in a frequency-dependent manner (Fig. 8h, i). These results

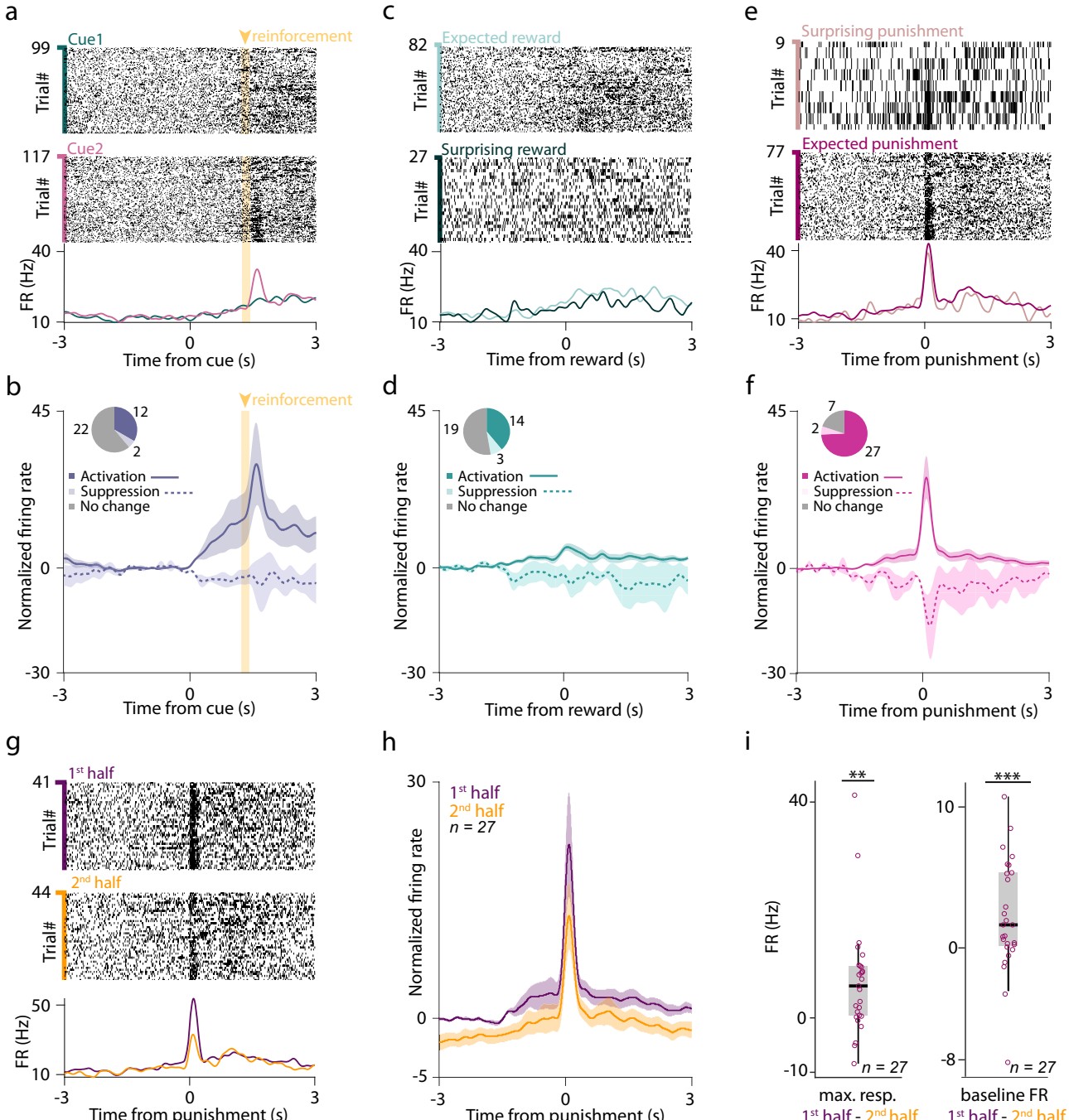

**Fig. 2 | BFPVNs of the HDB show strong responses to punishment. a** Raster plots and PETHs of the spike times aligned to cue onset of an example BFPVN. Trials are partitioned by the cue types (Cue 1 predicting likely reward / unlikely punishment, top, green; Cue 2 predicting likely punishment / unlikely reward, middle, purple). **b** Average PETH of BFPVN activity aligned to cue onset (activation, $n = 12$, solid line; inhibition, $n = 2$, dashed line; see inset for distribution of responses; errorshade, SEM). **c** Example PETH of the same BFPVN as in (**a**) aligned to reward delivery. Trials are partitioned into expected (following Cue 1) and surprising (following Cue 2) reward trials. **d** Average PETH of BFPVN activity aligned to reward delivery (activation, $n = 14$, solid line; inhibition, $n = 3$, dashed line; see inset for distribution of responses; errorshade, SEM). **e** Example PETH of the same BFPVN as in (**a**) aligned to punishment delivery. Trials are partitioned to surprising (following Cue 1) and expected (following Cue 2) punishment trials. **f** Average PETH of BFPVN activity

aligned to punishment delivery (activation, $n = 27$, solid line; inhibition, $n = 2$, dashed line; see inset for distribution of responses; errorshade, SEM). **g** Spike raster plots and PETH aligned to punishment delivery for an example BFPVN. Top, trials in the first half of the session; middle, trials in the second half of the session; bottom, PETH. **h** Average PETH of BFPVN activity aligned to punishment delivery ($n = 27$ punishment-activated BFPVNs). Purple, first half of the session; mustard, second half of the session (errorshade, SEM). **i** Left, difference in peak punishment response between the first and second half of the session (**$p < 0.01$, $p = 0.00458$, two-sided Wilcoxon signed-rank rest; $n = 27$ punishment-activated BFPVNs). Right, difference in baseline firing rate between the first and second half of the session (***$p < 0.001$, $p = 0.00071$, two-sided Wilcoxon signed-rank test; $n = 27$ punishment-activated BFPVNs). Boxes and whiskers show median, interquartile range and non-outlier range. Source data are provided as a Source Data file.

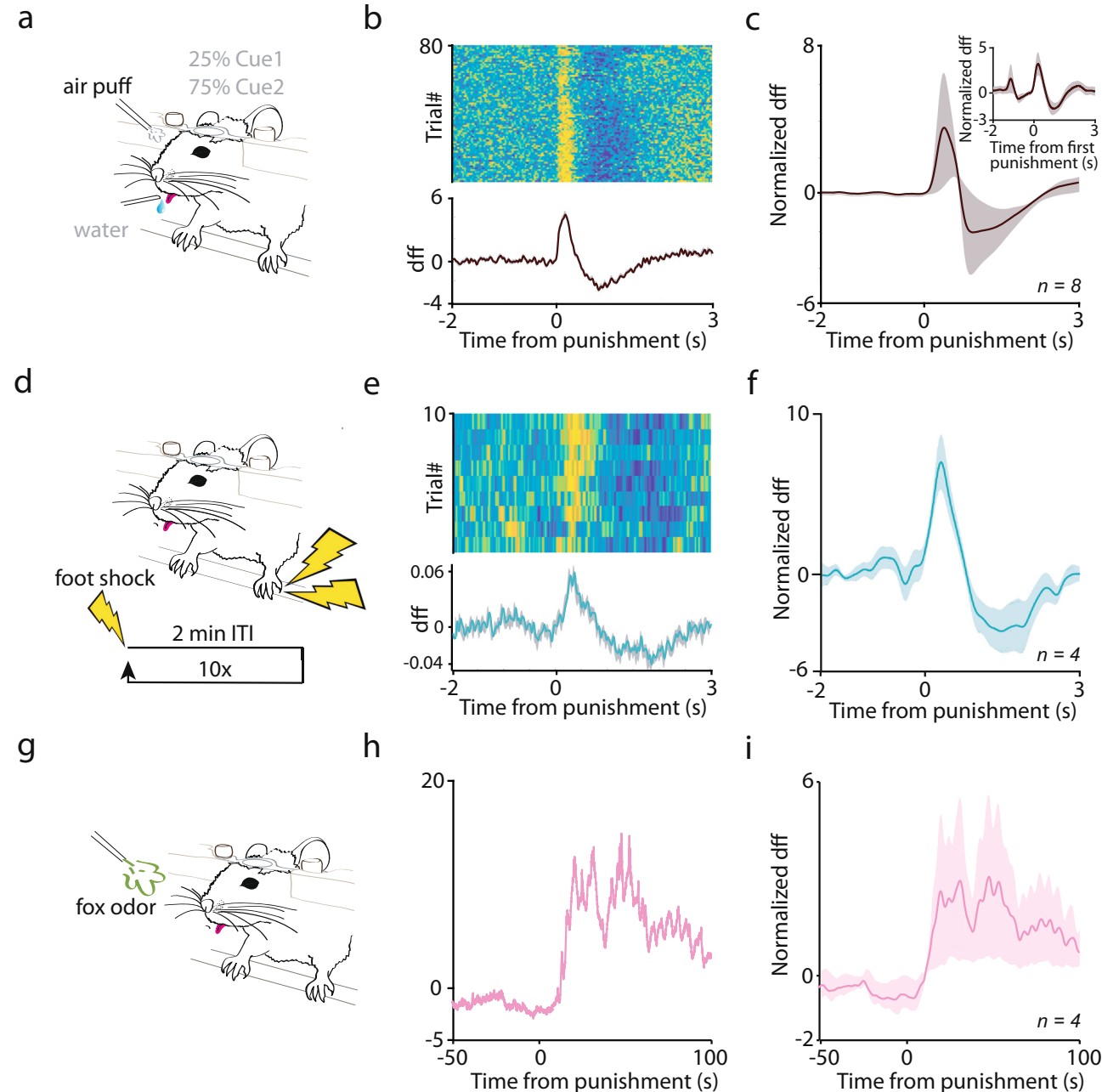

**Fig. 3 | BFPVNs respond to aversive stimuli of different modality (air puff, foot shock, fox odor) in naïve mice. a** Schematic diagram of introducing punishment during the Pavlovian conditioning task. **b** Example of a session when punishment was first introduced to the mouse. Top, PETH of individual trials; bottom, average (*n* = 80 trials; errorshade, SEM). **c** Average PETH of sessions when punishment was first introduced (*n* = 8 sessions from n = 8 mice). Inset, average BFPVN response to the first air puff presentation (*n* = 8 trials from n = 8 mice; errorshade, SEM). **d** Schematic diagram of foot shock presentation. **e** PETH aligned to foot shocks in an example session (*n* = 10 trials; errorshade, SEM). **f** Average PETH of foot shock response (*n* = 4 animals; 10 sessions each; errorshade, SEM). **g** Schematic diagram of fox odor presentation. **h** PETH aligned to fox odor presentation in an example session. **i** Average PETH of fox odor response in PVBFNs. (*n* = 4 animals; 1 session each; errorshade, SEM; note the longer time scale). PETHs were smoothed with a Gaussian kernel (width, 100 ms). (**a**), (**d**) and (**g**) panels were created using Mathis, M. (2020), Classical Conditioning Mouse, Zenodo, https://doi.org/10.5281/zenodo.3925907, under Creative Commons 4.0 license (https://creativecommons.org/licenses/by/4.0/). The original image was modified by removing illustrations of tubing and introducing additional legends and symbols.

suggest that BFPVN outputs to their prominent target regions might undergo reconfiguration depending on their firing rate, which varies with behavioral context.

## Discussion

We demonstrated that PV-expressing long-range inhibitory neurons in the HDB nucleus of the basal forebrain respond strongly and homogeneously to punishments, broadcast this activity to multiple downstream targets in the limbic system, and that this activity aids the learning of stimulus-outcome associations. These results suggest that basal forebrain PV-expressing neurons are important for learning from negative experiences.

The BF contains a substantial GABAergic population that is approximately five times larger than its cholinergic population according to stereological estimates in rodents[4,6]. Lesions of BF GABAergic neurons lead to learning deficits[22,40,41], and their gradual age-associated degeneration is linked to cognitive decline during healthy aging and Alzheimer disease[15–17], suggesting they have

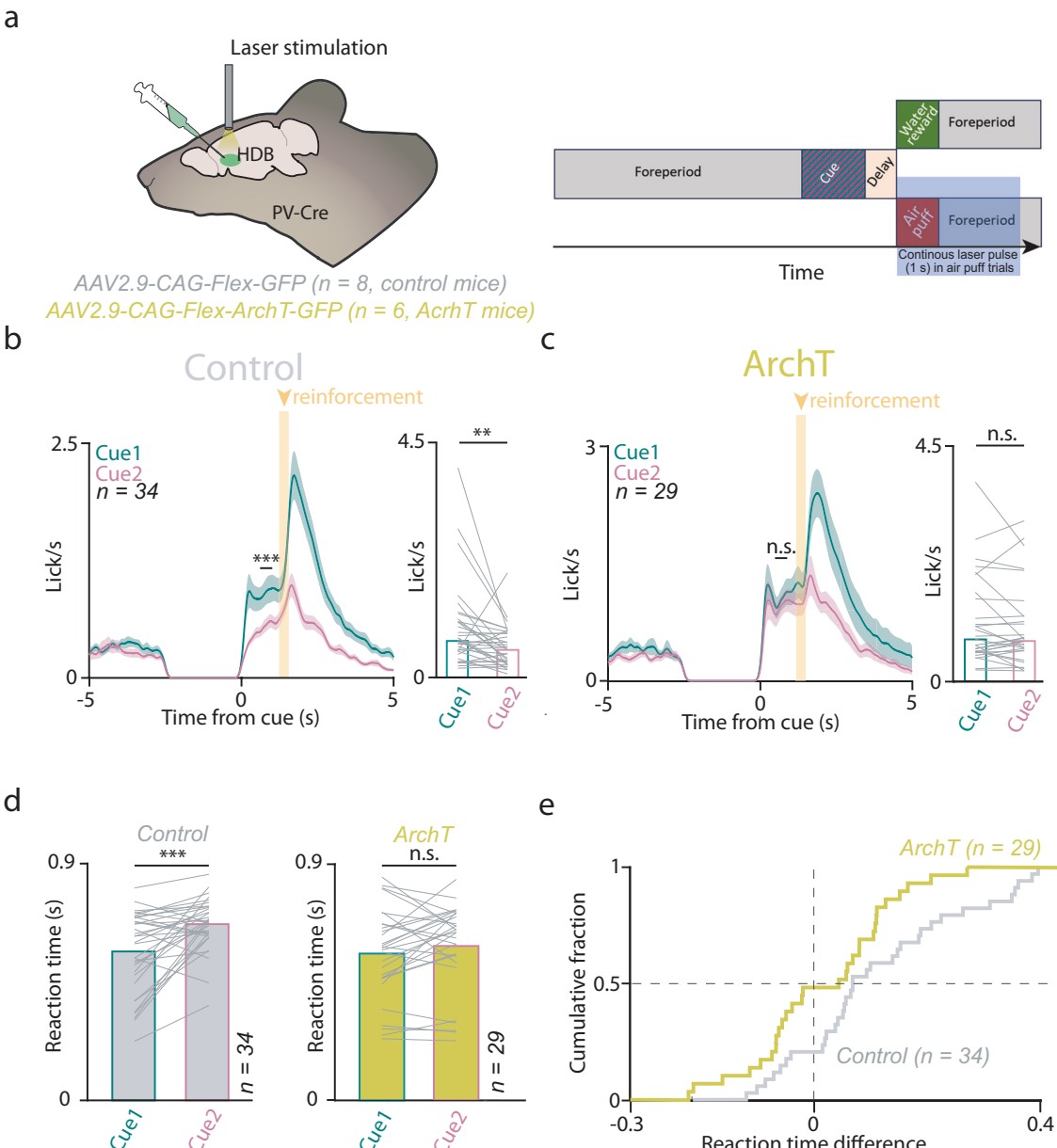

**Fig. 4 | Optogenetic suppression of HDB BFPVN punishment responses disrupts learning cue contingencies during Pavlovian conditioning. a** Left, schematic illustration of optogenetic suppression of BFPVNs in PV-Cre mice during air puff punishments. Created using Kennedy, A. (2020), Mouse brain silhouette, Zenodo, https://doi.org/10.5281/zenodo.3925919, under Creative Commons 4.0 license (https://creativecommons.org/licenses/by/4.0/). The original image was modified by adding labels and illustrations of a syringe and an optic fiber. Right, timeline of optogenetic suppression. **b** Left, average lick rate of control mice in the final training stage of the task (*n* = 34 sessions; errorshade, SEM). ***p < 0.001, ROC analysis with permutation test (see Methods). Right, bar graph showing mean lick rate before reinforcement (0.6–1.1 s window from cue onset, see ROC analysis in

Fig. S7). Lines correspond to individual sessions. **p < 0.01, p = 0.001517, two-sided Wilcoxon signed-rank test. **c**, Left, average lick rate of ArchT-expressing mice in the final training stage of the task (*n* = 29 sessions; errorshade, SEM). n.s., p > 0.05, ROC analysis. Right, bar graph showing mean lick rate in a 0.6–1.1 s window from cue onset. n.s., p > 0.05, p = 0.40513, two-sided Wilcoxon signed-rank test. **d** Left, reaction time difference between lick responses to Cue 1 and Cue 2 in control animals. ***p < 0.001, p = 0.000472, two-sided Wilcoxon signed-rank test. Right, same for the ArchT group. n.s., p > 0.05, p = 0.15668, two-sided Wilcoxon signed-rank test. **e** Cumulative histogram of reaction time differences (Cue 2–Cue 1) of ArchT (yellow, *n* = 29 sessions) and control (gray, *n* = 34 sessions) animals. Source data are provided as a Source Data file.

important functions in attention, learning and memory processes[6,10,42,43]. This inhibitory population contains a significant long-range projecting neuron pool, many of which express parvalbumin[7,8,42,44–46]. These projections include target areas implicated in cognitive processing, including frontal cortices and hippocampus[8,10,46].

We therefore asked whether the PV-expressing population of BF GABAergic projection neurons, the BFPVNs, are essential for

associative learning. We focused on the rostral portion of the horizontal nucleus of the diagonal band of Broca (HDB) and used bulk calcium recording via fiber photometry and tetrode recordings combined with optogenetic tagging to measure BFPVN activity in the HDB. We first tested whether BFPVNs responded to behaviorally salient events during classical conditioning, while mice learned to associate previously neutral pure tone stimuli (CS) with water rewards and air puff punishments (US). Previous studies showed that a bursting

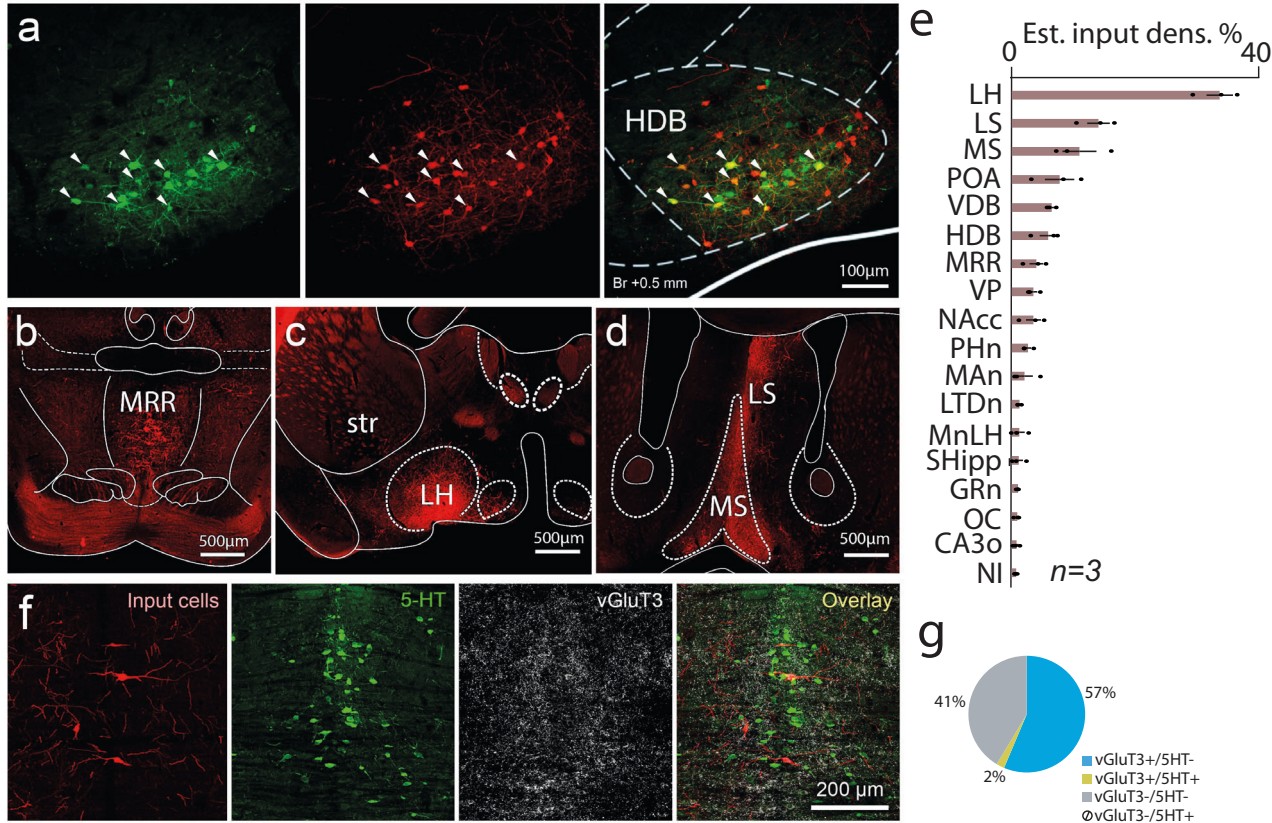

**Fig. 5 | BFPVNs of the HDB receive inputs from aversion-coding nuclei.**
**a** Retrograde tracing, using pseudotyped rabies virus. Green, helper virus (left); red, pseudotyped rabies virus (middle); yellow, colocalizaton (right). Starter cells in the HDB are indicated by white arrowheads. **b** Coronal fluoromicrograph showing input cells in the MRR (red fluorescence). **c** Coronal fluoromicrograph showing input cells in the lateral hypothalamus (red fluorescence). **d** Coronal fluoromicrograph showing input cells in the medial septum (red fluorescence). **e** Estimated input density (% of all input cells) in different input brain regions. Bars and error-bars indicate the mean ± SEM from n = 3 mice. **f** Immunohistochemical staining of input neurons (red), serotonergic neurons (5-HT, green) and vGluT3+ glutamatergic neurons (vGluT3, far red visualized in white) in the MRR.
**g** Proportion of neurons in the MRR that target BFPVNs. LH lateral hypothalamus, LS lateral septum, MS medial septum, POA preoptic area, VDB vertical limb of the diagonal band of Broca, HDB horizontal limb of the diagonal band of Broca, MRR median raphe region, VP ventral pallidum, NAcc nucleus accumbens, PHn posterior hypothalamic nucleus, MAn medial amygdaloid nucleus, LTDn laterodorsal tegmental nucleus, MnLH magnocellular nucleus of the lateral hypothalamus, SHipp septohippocampal nucleus, GRn gigantocellular reticular nucleus, OC orbital cortex, CA3o CA3 stratum oriens, NI nucleus incertus. Source data are provided as a Source Data file.

population of BF non-cholinergic neurons responded to behaviorally relevant events irrespective of their valence during associative learning[23,24]. We also found that BFPVNs responded to salient events including cue tones, reward and punishment; however, in contrast with Lin and colleagues, we demonstrated a preference of HDB BFPVNs towards responding to the aversive US, with only weaker and less consistent responses to CS and reward, suggesting that HDB BFPVNs constitute a specialized subpopulation of non-cholinergic BF neurons. We also found that BFPVNs markedly differed from cholinergic BF neurons, the BFCNs, in their responses: our previous studies revealed outcome prediction coding of BFCNs both following CS and US[25,34–36], absent in HDB BFPVNs. We also found that HDB BFPVNs were responsive to electrical shock and predator odor, demonstrating that BFPVN responses to aversive stimuli are multimodal.

BFPVN responses to air puffs showed a gradual decrease during the recording sessions. This could indicate that the aversive quality of the air puffs decreased with repeated exposure; however, other potential contributing sources like decreasing motivation, stimulus novelty, and fatigue cannot be excluded.

An unbiased clustering of HDB neuron responses to behaviorally relevant events in the probabilistic Pavlovian conditioning task grouped most BFPVNs in two clusters that together covered 48% of the recorded HDB cells. Non-tagged neurons in these clusters could

represent both undetected BFPVNs due to limitations of viral infection and photoactivation, and, based on cluster size, likely also other HDB neuron types activated after reinforcement[4,6]. A smaller cluster contained cells that showed suppression of firing after punishment and to a lesser extent after reward. These responses mirrored BFPVN response profiles, suggesting that some of these neurons might be locally inhibited by the GABAergic BFPVNs. Such connections were found to be rare by circuit mapping studies[6,12], and in accordance, this group contained only 14% of HDB neurons.

The basal forebrain contains another prominent GABAergic cell type that expresses somatostatin (SOM). These BFSOMNs inhibit BFCNs, BFPVNs and BF glutamatergic neurons[12] via GABAergic synapses. In addition, SOM itself presynaptically inhibits glutamate and GABA release onto BFCNs[2]. BFSOMNs receive excitatory inputs from local glutamatergic neurons as well as BFCNs via nicotinic ACh receptors, while muscarinic receptors convey slower hyperpolarizing inputs[6]. Many of these cells are sleep-active and sleep promoting, suggesting that BFSOMNs potently suppress all wake-promoting BF cell types during non-REM sleep[12]. At the same time, specific activity of BFSOMNs during behavior was not known. By performing bulk calcium recordings of these neurons, we found that BFSOMNs responded to rewards, punishments, and reward-predicting auditory cues. This activity pattern differentiated them from BFPVNs that were most active after punishments and was

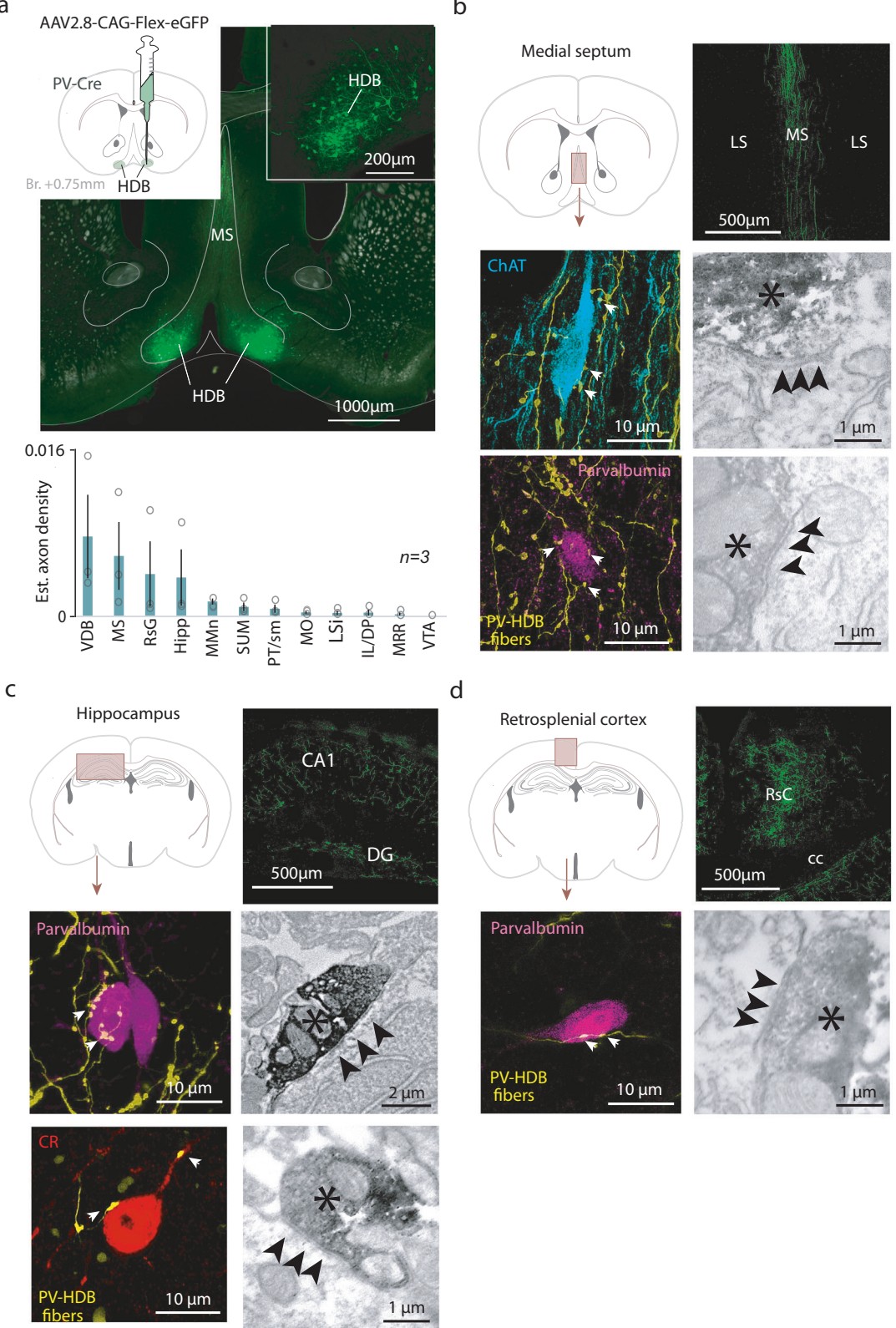

remarkably similar to BFCN calcium responses[35]. However, no overlap was found between SOM- and ChAT-expressing neurons in the HDB (Fig. S4c), as reported before[6]. Additionally, BFSOMNs showed higher activity after surprising than after expected rewards, similar to BFCNs and unlike BFPVNs. However, this higher activity occurred on the background of higher calcium levels at times of surprising reward delivery, due to a suppression of the calcium signal following the activation induced by the

reward-predicting cue (not observed for BFCNs[35]), complicating the interpretation of this difference.

Thus, BFSOMNs responded to the same behaviorally salient events as BFPVNs but differed in relative response magnitudes (e.g. large reward responses for BFSOMNs) and coding properties (differential response to surprising and expected outcomes in BFSOMNs). We propose that BFSOMNs might provide negative feedback onto

**Fig. 6 | BFPVNs of the HDB innervate the limbic system. a** Top left, schematic illustration of cell-type specific anterograde tracer virus injection into the HDB. Top right and middle, fluorescent image of the injection site (green, eGFP). Bottom, normalized BFPVN axon density in target brain areas. Bars and errorbars indicate the mean ± SEM from n = 3 mice. **b** Top left, schematic illustration of the medial septum on a coronal atlas image. Top right, fluoromicrograph of labeled axons in the MS (repeated in n = 3 mice). Middle and bottom left, fluorescent images of ChAT+ and PV+ target neurons in the medial septum (white arrowheads indicate plausible contact sites). Right, electronmicrographs of synaptic contacts established by BFPVNs with the target cell types shown in the left panels (different examples). HDB BFPVN axons are labeled by DAB precipitate (black asterisk); black arrows indicate synaptic contacts. **c**, Top left, schematic illustration of the hippocampus on a coronal atlas image. Top right, fluoromicrograph of labeled axons in the CA1 (repeated in n = 3 mice). Middle and bottom left, fluorescent images of CR+ and PV+ target neurons in the hippocampus. Middle and bottom right, electronmicrographs of synaptic contacts established by BFPVNs with the target cell types shown in the left panels. **d** Top left, schematic illustration of the retrosplenial cortex (RSC) on a coronal atlas image. Top right, fluoromicrograph of labeled axons in the RSC (repeated in n = 3 mice). Bottom left, fluorescent images of a PV+ target neuron in the retrosplenial cortex. Bottom right, electronmicrograph of a synaptic contact established by BFPVNs with the target cell type shown on the left. VDB ventral limb of the diagonal band of Broca, MS medial septum, RsG retrosplenial cortex, granular part, Hipp hippocampus, MMn meidal mamillary nucleus, SUM supramamillary nucleus, PT/sm paratenial thalamic nucleus/stria medullaris, MO medial orbital cortex, LSi lateral septum, intermediate part, IL/DP inflalimbic cortex/dorsal peduncular cortex, MRR median raphe region, VTA ventral tegmental area. Source data are provided as a Source Data file.

BFCNs, both inhibiting their outputs (via GABA release) and decoupling them from their inputs (via SOM release), thereby limiting the duration of their activation. Previous recording studies showing fast and precise activation of BFCNs support this notion[25,35,36], suggesting that a fast bottom-up activation of the cholinergic system might recruit negative BFSOMN feedback via nicotinic receptors[6].

Information about aversive outcomes has multiple relevance for the animals, likely involving partly overlapping, but partly divergent circuits. First, aversive outcomes typically (but not mandatorily, see ref. 47) evoke avoidance behavior, engaging effector circuits and eventually muscles. A number of nuclei and cell types (mostly glutamatergic) have been shown to be involved in active avoidance[37,48–51]. Second, animals have to learn from aversive information, e.g. form stimulus-outcome associations that will then be stored in memory[50]. Third, aversive information also leads to arousal, attention and increased vigilance. These are different processes induced by aversive sensory information; therefore, it is expected that they be mediated by at least partially separable circuits. For example, basal forebrain cholinergic neurons respond rapidly and reliably to aversive stimuli including shocks and air puffs, but their stimulation evokes neither avoidance nor approach[25,52]. Indeed, they are thought to contribute to the learning aspects of outcomes by controlling cortical plasticity[25,53–56]. Being part of the same anatomical structure, BFPVNs are likely also involved in cognitive processing rather than direct motor effector functions. In line with this, they were hypothesized to contribute to arousal on both slower and faster timescales[6,10,12,13]. The lack of place avoidance after photostimulation of BFPVNs combined with impaired learning caused by their photoinhibition confirmed this hypothesis. We propose that HDB BFPVNs might specifically increase attention for aversive learning, thus mediating associative learning processes through increasing cortical excitability at specific target areas, probably by disinhibition.

Optogenetic inhibition of HDB BFPVNs during air puff punishments prevented mice from forming differential representation of CS cues based on their valence, demonstrated by a lack of differential anticipatory lick responses in optogenetically manipulated mice within the course of training. This lack of difference in CS-elicited licking was apparently due to an impaired inhibition of licking after Cue 2 that predicted likely punishment, in line with a previous study in which ibotenic acid injected to the BF of rats, preferentially destroying non-cholinergic neurons, lead to an increased false alarm rate in a visual detection task[18].

What may be the input that conveys aversive information to HDB BFPVNs? To address this question, we mapped the input neurons to HDB BFPVNs by mono-transsynaptic retrograde tracing using rabies vectors. We found that the input patterns of HDB BFPVNs only partially overlapped with those of a broader PV-expressing population[8]; most importantly, the lateral hypothalamus showed the largest density of input neurons, known to transmit nociceptive information that it receives via direct monosynaptic connections from nociceptive

neurons of the spinal cord and periaqueductal gray[57–59]. HDB BFPVNs receive inputs from different BF nuclei including the medial septum and the ventral pallidum, which may also transmit punishment-related signals[2,25,60–62]. Furthermore, the median raphe was also sending monosynaptic inputs to BFPVNs, raising the possibility of a direct brainstem source of aversive information[37,63,64]. This was supported by our finding that MRR input cells consisted mostly of vGluT2- and vGluT3-expressing glutamatergic neurons, known to play important roles in the brainstem control of negative experience[37,39]. Importantly, the remaining major inputs to HDB BFPVNs were the lateral septum and the nucleus accumbens, thought to be important regulators of affect[65]. The potential convergence of aversive/affective information on BFPVNs revealed by rabies-based input mapping suggests that these neurons may act as an integrating and processing hub for aversive information. Nonetheless, we cannot rule out the possibility that one of these inputs dominates over the others, and we have limited knowledge on whether other HDB neurons participate in this process through local connections onto BFPVNs. Of note, lateral hypothalamic inputs dominated over striatal inputs to HDB BFPVNs, while these two sources were balanced in a broader PV-expressing population[8], which may underlie the robust punishment responses of HDB BFPVNs in comparison to reward-related activity.

We tested by anterograde tracing, fluorescence and electron microscopy to what target areas and neurons this aversive information was transmitted. HDB BFPVNs were projecting to neighboring BF nuclei, the vertical limb of the diagonal band of Broca and the medial septum (MS-VDB), where they synapsed on cholinergic and PV-expressing GABAergic neurons. They also sent dense projections to the hippocampus, where they formed synaptic contacts on PV-expressing and calretinin-expressing hippocampal interneurons. Of note, this innervation pattern was similar to what had been found for the septohippocampal GABAergic projection[44,66]. A third major target of HDB BFPVNs was the retrosplenial cortex, interfacing multimodal sensory and limbic information[67,68]. Previous studies suggested that long-range projecting GABAergic BF neurons may be mostly disinhibitory[69–74], which we confirmed in the CA1 and in the retrosplenial cortex, where we demonstrated HDB BFPVN synapses on PV-expressing interneurons; however, potential contacts onto pyramidal neurons could not be reliably tested. Further targets of HDB BFPVNs included other limbic structures like the mammillary and supramammillary nuclei and the paratenial thalamic nucleus, and to a lesser extent areas of the frontal cortex. This innervation pattern showed considerable differences from the projection pattern of a broader PV-expressing BF population[8], while it was remarkably similar to subcortical projections of HDB GABAergic neurons[7], suggesting that the PV-expressing population might provide a substantial fraction of the BF inhibitory projections[6,10,75], while it may show strong topography, similar to other BF cell types[2,76,77]. We note that while we found relatively sparse connections to frontal areas like medial orbital and infralimbic cortices, caudal portion of the HDB may show denser

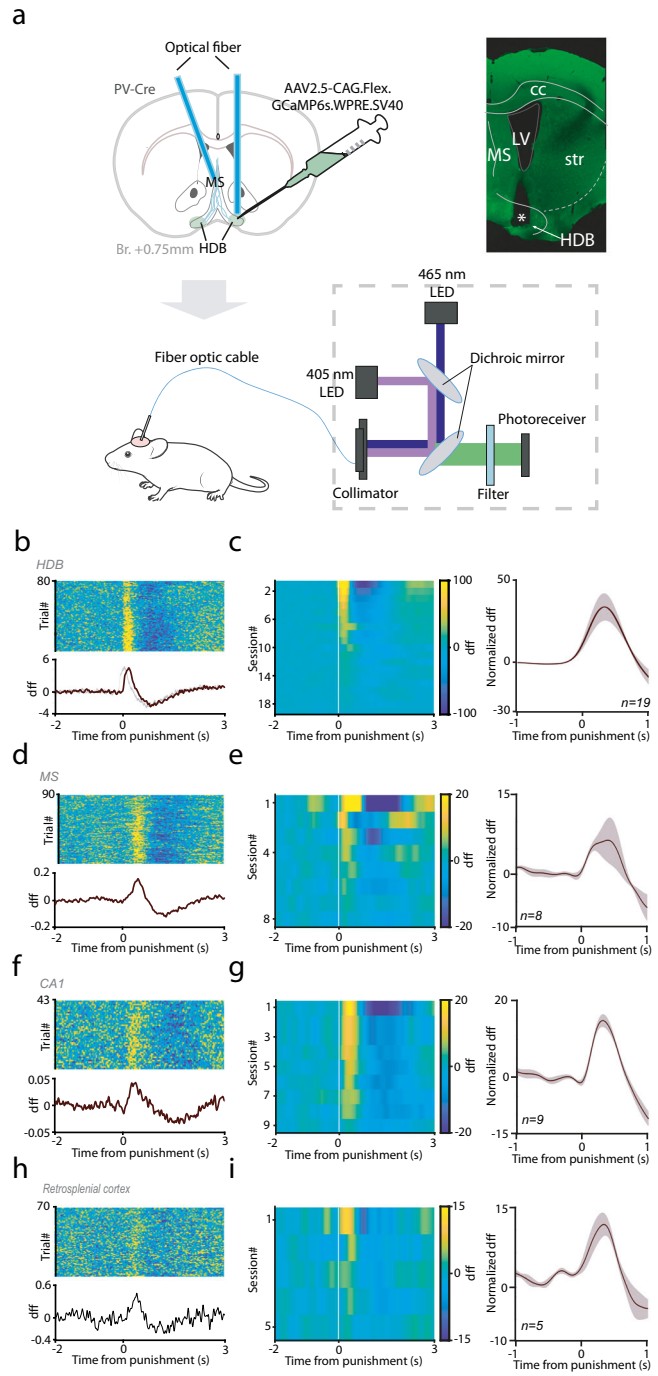

**Fig. 7 | HDB BFPVNs broadcast negative feedback information to multiple limbic target regions. a** Schematic illustration of bulk calcium measurements of BFVN activity in the HDB and in the main target areas. Top left, injection of GCaMP6s-expressing viral vector; top right, fluoromicrograph of an optical fiber track (green, GCaMP6s; asterisk, tip of the optical fiber). Bottom, schematic of bulk calcium recording. Created using Tyler, E., & Kravitz, L. (2020), Mouse, Zenodo, https://doi.org/10.5281/zenodo.3925901, under Creative Commons 4.0 license (https://creativecommons.org/licenses/by/4.0/). The original image was modified by adding illustrations of an optic cable and photometry recording system. **b**, **c** Fiber photometry recordings of HDB BFPVN somata. **b** Example session recorded in the HDB. Top, single trial dff traces; bottom, average PETH aligned to air puff punishments (n = 80 trials; errorshade, SEM). **c** Left, all single sessions of HDB recordings, sorted according to descending amplitudes. Right, mean PETH of all sessions (n = 19; errorshade, SEM). **d–i** Fiber photometry recordings of HDB BFPVN projections. **d** Example session recorded in the medial septum. Top, single trial dff traces; bottom, average PETH aligned to air puff punishments (n = 90 trials; errorshade, SEM). **e** Left, all single sessions of medial septum recordings, sorted according to descending amplitudes. Right, mean PETH of all sessions (n = 8; errorshade, SEM). **f** Example session recorded in the hippocampal CA1 region. Top, single trial dff traces; bottom, average PETH aligned to air puff punishments (n = 43 trials; errorshade, SEM). **g** Left, all single sessions of CA1 recordings, sorted according to descending amplitudes. Right, mean PETH of all sessions (n = 9; errorshade, SEM). **h** Example session recorded in the retrosplenial cortex. Top, single trial dff traces; bottom, average PETH aligned to air puff punishments (n = 70 trials; errorshade, SEM). **i** Left, all single sessions of retrosplenial cortex recordings, sorted according to descending amplitudes. Right, mean PETH of all sessions (n = 5; errorshade, SEM). PETHs were smoothed with a Gaussian kernel (width, 100 ms).

At the same time, in vitro electrophysiology recordings revealed substantial differences in processing inputs from BFPVNs in target neurons: inhibitory synaptic responses of CA1 hippocampal interneurons were relatively large and fast and showed prominent short-term depression, while inputs on RSC neurons were smaller in amplitude, had longer latency to peak as well as rise time, and showed frequency-dependent short-term facilitation. In the MS, where BFPVNs synapsed on PV-expressing and cholinergic neurons, the synaptic properties showed intermediate values. Thus, the impact of BFPVN activity across target areas could dynamically change with varying behavioral and brain states.

Although theoretically possible, it is unlikely that BFPVNs would exert their effects through local connections within the HDB, since they do not innervate local BFCNs or BFSOMNs, and provide only weak and sparse connections to local glutamatergic neurons[6,12]. In line with this, Kim et al. found that the BFPVNs impact cortical gamma oscillations independent of BFCNs[10].

BFPVNs have been shown to control cortical gamma oscillations[6,9–11] and were suggested to participate in mediating arousal at different timescales, including brief and rapidly changing microarousals[6,9,12]. Relatedly, they were proposed to control changing levels of attention through cortical activation[6,9,25]. BFPVNs were also suggested to activate the default mode network, which is likely responsible for maintaining global oscillatory activity necessary for higher cognitive functions and conscious experience in humans, determining the significance of which will require future studies[85,86]. Combining these and our results, we speculate that BFPVNs may rapidly disinhibit limbic areas upon aversive stimuli to facilitate learning from negative experience. Thus, at least for aversive stimuli, BFPVNs might be the physical substrate of the 'attention for learning' concept[9,87].

## Methods

### Animals

Adult (over two months old) male Pvalb-IRES-Cre (PV-IRES-Cre or PV-Cre; n = 60, The Jackson Laboratory, RRID: IMSR_JAX:017320) and SOM-IRES-Cre (also known as Sst-IRES-Cre; n = 4, The Jackson Laboratory, RRID: IMSR_JAX:013044) mice were used for recording,

frontal cortical projections[10]. Importantly, the innervation pattern revealed by the tracing indicates a broad targeting of the limbic system by HDB BFPVNs, multiple areas of which were shown to be necessary for learning from aversive experience[49,78–84].

Calcium recording of HDB BFPVN projections performed at three major termination zones, the medial septum, the CA1 of the hippocampus and the retrosplenial cortex revealed that all HDB BFPVN projections exhibited similar response properties to what was observed at the cell body level. Specifically, all three projections showed prominent activation by air puff punishments. Although some quantitative differences could be observed, it was not possible to dissect whether it was due to differences in axon density or differential response strength at the projection level. Altogether, the qualitatively similar response profiles suggest a broadcast function of HDB BFPVNs, sending similar message to broad areas of the limbic system.

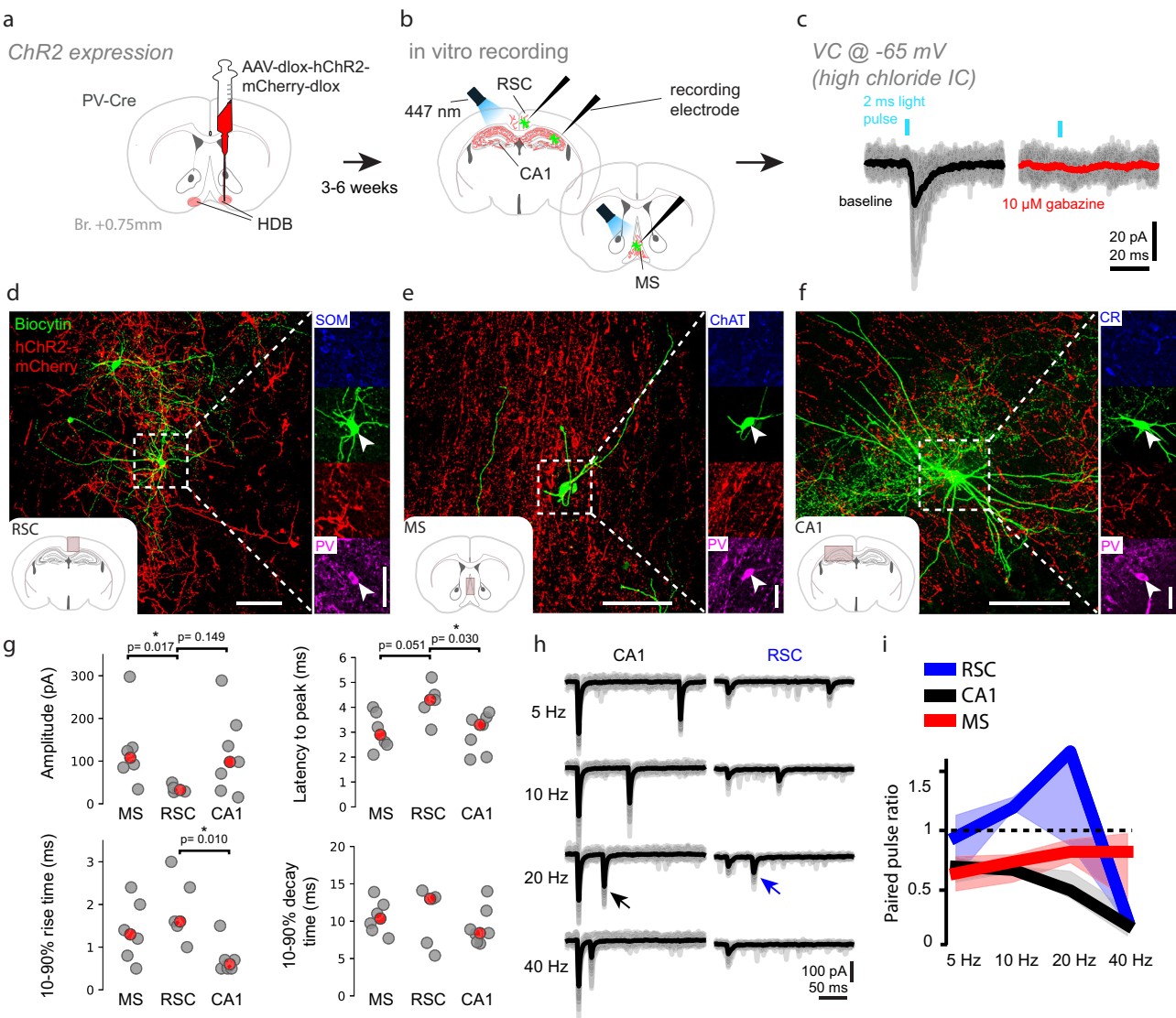

**Fig. 8 | Functional synaptic contacts of BFPVNs show target area specific short-term plasticity. a** PV-Cre mice (*n* = 7) were injected with AAV to express ChR2 in BFPVN axonal projections. **b** Acute slice electrophysiology recordings were performed from the retrosplenial cortex (RSC), CA1 area of the dorsal hippocampus and the medial septum (MS). **c** Left, representative example of postsynaptic inhibitory currents recorded from a neuron in response to the optogenetic stimulation of BFPVN fibers. Right, the inhibitory response was eliminated in the presence of GABA_A blocker gabazine (10 µM). **d–f** Representative confocal images of responsive cells from the RSC, MS, and CA1, respectively (green, recorded cells; red, BFPVN axonal projections; scale bars: 100, 30 µm). Post-hoc immunohistochemistry shows functional contacts on PV+ cells in RSC, CA1 and MS. (*n* = 4/7 patched cells were PV+ in the CA1; *n* = 2/5 in the MS; *n* = 2/5 in the RSC). **g** Top left, optogenetically evoked IPSCs were significantly smaller on RSC neurons (*n* = 5) compared to MS neurons (*n* = 6) and showed a tendentious difference compared to CA1

neurons (*n* = 7; RSC vs. MS, *p* = 0.017; RSC vs. CA1, *p* = 0.149). Latency to peak was higher in the RSC compared to the other two regions (RSC vs. MS, *p* = 0.051; RSC vs. CA1, *p* = 0.030). In line with these, 10–90% rise-time was shortest in the CA1 (RSC vs. CA1, *p* = 0.010). We found no differences in 90–10% decay time. Statistical comparisons were performed using two-sided Mann–Whitney *U* test. **h** We delivered two consecutive light pulses at different frequencies to calculate paired-pulse ratios (PPR). Responses from representative CA1 and RSC neurons are shown. Note the prominent short-term synaptic depression of the BFPVN input onto the CA1 cell (black arrow), and the short-term facilitation up to 20 Hz of the BFPVN input onto the RSC cell (blue arrow). **i** Summary data of PPR at different frequencies in the three brain areas (RSC vs. CA1 @ 10 Hz, *p* = 0.008, RSC vs. MS @ 10 Hz, *p* = 0.030, RSC vs. CA1 @ 20 Hz, *p* = 0.051, two-sided Mann–Whitney *U* test). Source data are provided as a Source Data file.

optogenetic activation and inhibition, bulk calcium recording and for immunohistochemistry, and 30-60 days old PV-IRES-Cre mice were used for acute slice electrophysiology (*n* = 7, males) according to the regulations of the Hungarian Act of Animal Care and Experimentation (1998; XXVIII, section 243/1998, renewed in 40/2013) in accordance with the European Directive 86/609/CEE and modified according to the Directive 2010/63/EU. Experimental procedures were reviewed and approved by the Animal Welfare Committee of the Institute of Experimental Medicine, Budapest and by the Committee for Scientific Ethics of Animal Research of the National Food Chain Safety Office of

Hungary (PE/EA/675-4/2016; PE/EA/1212-5/2017; PE/EA/864-7/2019; PE/EA/1003-7/2021). Animals were housed individually in 36 × 20 × 15 cm cages under a standard 12-h light-dark cycle (lights on at 8 a.m.) with food available ad libitum. During behavioral training, mice were water-restricted to 1 ml per day. Temperature and humidity were kept at 21 ± 1 °C and 50–60%, respectively.

**Tetrode implantation surgery**
Mice were implanted with microdrives housing 8 tetrodes and a 50 µm core optic fiber[25,88,89]. Mice were anesthetized with a ketamine-xylazine

solution (83 mg/kg ketamine and 17 mg/kg xylazine, dissolved in 0.9% saline). The scalp was shaved and disinfected with Betadine, the skin and subcutaneous tissues were topically anesthetized with Lidocaine spray, and mice were placed in a stereotaxic frame (Kopf Instruments). Eyes were protected with eye ointment (Corneregel, Bausch & Lomb). The skin was opened with a single sagittal incision made by a surgical scalpel, the skull was cleaned, and a craniotomy was drilled above the horizontal limb of the diagonal band of Broca (HDB, antero-posterior 0.75 mm, lateral 0.60 mm; $n = 4$). Virus injection (AAV2/5.EF1a.-Dio.hChR2(H134R)-eYFP.WPRE.hGH, Addgene, titer $\geq 1 \times 10^{13}$ vg/mL; HDB, dorso-ventral 5.00 and 4.70 mm, 300 nl at each depth) and drive implantation was performed using the stereotaxic frame and a programmable nanoliter injector (Drummond Nanoject III). Ground and reference electrodes were implanted to the bilateral parietal cortex. The microdrive was secured in place using dental cement (Lang Dental). When the dental cement was cured, mice were transferred from the stereotaxic frame into their homecage and received analgesics (Buprenorphine, 0.1 mg/kg), and local antibiotics (Gentamycin). Mice were closely monitored after surgery and were allowed 10-14 days of recovery before starting behavioral training.

### Anterograde tracing surgery
PV-IRES-Cre mice were injected with AAV2/8.CAG.Flex.eGFP (Addgene, titer $\geq 1 \times 10^{13}$ vg/mL) anterograde tracing virus in the HDB (antero-posterior, 0.75 mm; lateral, 0.60 mm; dorso-ventral, 5.00 mm and 4.9 mm; 50 nl each side, injecting 25–25 nl each depth) using standard stereotaxic surgery techniques described in the previous section. Following bilateral virus injection, craniotomies were closed with Vaseline, and scalp skin was sutured using standard surgical needle and thread (Dafilon). Postoperative care was applied as described above. The virus injection was followed by a 4-week virus expression period. Mice were then transcardially perfused (as described in details below) and their brains were processed for further immunohistology experiments.

### Retrograde tracing surgery
PV-IRES-Cre mice were anesthetized with 2% isoflurane followed by an intraperitoneal injection of a ketamine-xylazine anesthetic mixture (83 mg/kg ketamine and 17 mg/kg xylazine, dissolved in 0.9% saline), then mounted in a stereotaxic frame (David Kopf Instruments). After exposing the skull surface, 20 nl of the Cre-dependent helper virus was injected into the HDB (bilateral, $n = 2$; unilateral, $n = 1$) using a Nanoliter 2010 microinjection pump (World Precision Instruments). The injection coordinates were defined by a stereotaxic atlas (Paxinos and Franklin, 2012) and were the following given in mm for the anteroposterior, mediolateral and dorsoventral axes, respectively: +0.75 (from Bregma), ±0.60 (from Bregma, unilateral in case of Mouse 3), −5.0 (from brain surface). After 3 weeks of survival, mice were injected with 20 nl of the G protein-deleted EnvA-pseudotyped rabies virus at the same coordinates. After 9 days of survival, mice were prepared for perfusion.

### Optical fiber implantation surgery for optogenetic activation, inhibition and fiber photometry recordings
PV-IRES-Cre or SOM-IRES-Cre (also known as Sst-IRES-Cre) mice were implanted using standard stereotactic surgery techniques described in the previous section. Following virus injection (for optogenetic activation: AAV2/5.EF1a.Dio.hChR2(H134R)-eYFP.W-PRE.hGH, Addgene, titer $\geq 1 \times 10^{13}$ vg/mL or AAV2/5.EF1a-eYFP.W-PRE.hGH, Addgene, titer $\geq 7 \times 10^{12}$ vg/mL as control virus, 70 nl each side; for optogenetic inhibition: AAV2.9-CAG-Flex-ArchT-GFP, Addgene, titer $\geq 1 \times 10^{13}$ vg/mL or AAV2.9-CAG-Flex-GFP, Addgene, titer $\geq 1 \times 10^{13}$ vg/mL as control virus, 100 nl each side; for fiber photometry: AAV2/9.CAG.Flex.GCaMP6s.WPRE.SV40, Addgene, titer $\geq 1 \times 10^{13}$ vg/mL; HDB, antero-posterior 0.75 mm, lateral

0.60 mm; dorso-ventral 5.00 and 4.7 mm). Following virus injection, optical fibers were implanted (for optogenetic activation: 200 μm core fiber was implanted unilaterally into HDB, antero-posterior 0.75 mm, lateral 0.60 mm; for optogenetic inhibition: 400 μm core fibers were implanted bilaterally in 0- and 20-degree angle into HDB; for fiber photometry: 400 μm core fibers were implanted into HDB and MS, antero-posterior 0.75 mm, lateral 1.2 mm, 20-degrees lateral angle; CA1 region of the hippocampus, antero-posterior −2.3 mm, lateral 1.4 mm, dorso-ventral 1.2 mm; retrospelnial cortex, antero-posterior −2.3 mm, lateral 0.5 mm, dorso-ventral 0.2 mm respectively). Mice received post-surgery care as described above in the *Tetrode implantation surgery* section and allowed 10–14 days of recovery before starting behavioral training.

### Probabilistic Pavlovian conditioning
PV-IRES-Cre or SOM-IRES-Cre mice were trained on an auditory Pavlovian conditioning task[26] in a head-fixed behavioral setup that allowed millisecond precision of stimulus and reinforcement delivery[90]. On the first day of training, water restricted mice[26] were head-fixed and given access to water reward whenever they licked a waterspout. Individual licks were detected by the animal's tongue breaking an infrared photobeam. The following day, a 50 dB pure tone cue of one second duration was introduced that predicted likely reward (5 μl of water). After each cue presentation, water reward was delivered with 0.8 probability with a 200–400 ms delay, while the rest of the outcomes were omissions. The next trial started after the animal stopped licking for at least 1.5 s. The stimulus was preceded by a 1–4 s foreperiod according to a truncated exponential distribution, in order to prevent temporal expectation of stimulus delivery[91]. If the mouse licked in the foreperiod, the trial was restarted. We used the open source Bpod behavioral control system (Sanworks LLC, US) for operating the task.

Next, a second pure tone cue of well-separated pitch was introduced that predicted low probability reward (0.25). Next, air puff punishment (200 ms, 15 psi) was introduced in the following session with the final outcome contingencies (likely reward trials, 80% reward, 10% punishment, 10% omission; likely punishment trials, 25% reward, 65% punishment, 10% omission). These contingencies reflect careful calibration to keep mice motivated for the task. Different trial types (likely reward and likely punishment) and outcomes (water reward, air puff punishment and omission) were presented in a pseudorandomized order following the described outcome contingencies. Mice learned the task in approximately one week and consistently demonstrated reward anticipation by differential lick rate in response to the cues from the second week.

Auditory stimuli were calibrated using a precision electret condenser microphone (EMM-6, Daytonaudio) connected to a pre-amplifier digital converter (AudioBox iOne, PreSonus) and sound pressure levels were measured by the TrueRTA software[90]. Cue tone intensities were well above the hearing threshold for both sounds (20-30 dB for 4 and 12 kHz pure tones) but not as loud as to cause a startle reflex (around 70 dB)[25,92]. Mice in this study underwent the same training protocol for consistency; however, we have reported that behavioral performance of the task did not depend on the identity (frequencies) of the conditioned stimuli (Figure S1 in ref. 35).

The aversive quality of air puffs depends on the exact experimental settings. We applied 200 ms long puffs at 15 psi pressure (within the range of parameters used for eyeblink conditioning[93]). We demonstrated that mice consistently choose water without air puff over water combined with air puff, showing that air puffs are aversive under these circumstances (see Figs. 2C and 2D in ref. 25).

### Optogenetic inhibition of BFPVNs
A set of PV-IRES-Cre animals injected with AAV2.9-CAG-Flex-ArchT-GFP (Addgene, titer $\geq 1 \times 10^{13}$ vg/mL, $n = 6$, ArchT group) or AAV2.9-CAG-

Flex-GFP (Addgene, titer $\geq 1 \times 10^{13}$ vg/mL, $n = 8$, Control group) were trained for a two-week period on the probabilistic Pavlovian conditioning task according to the training protocol described above[26]. Starting at the time of punishment presentation, a one-second-long yellow laser pulse (593 nm wavelength, Sanctity Laser) was delivered to the HDB bilaterally via optical fibers (400 μm core) to achieve optogenetic silencing of BFPVNs, introduced already at the first air puff delivery. Anticipatory lick rate (licking between cue onset and reinforcement delivery) was recorded in each (ArchT and Control) group.

## Conditioned place aversion task
PV-IRES-Cre animals injected with AAV2/5.EF1a.Dio.hChR2(H134R)-eYFP.WPRE.hGH (Addgene, titer $\geq 1 \times 10^{13}$ vg/mL, $n = 8$, ChR group) or AAV2/5.EF1a-eYFP.WPRE.hGH (Addgene, titer $\geq 7 \times 10^{12}$ vg/mL, $n = 10$, Control group) were placed in an arena with two sides, which were separated by a wall but connected with an open gate on the wall. The two sides of the arena were marked by different wall patterns (dotty or striped). In the arena, a center part (one third of the size of the arena) and a peripheral part was differentiated. The conditioning protocol took two days; on the first day, animals were placed into the chamber and were allowed to move freely for 5 min (habituation phase). The next day (test phase), mice were place into the chamber again and received 20 Hz laser light stimulation (473 nm wavelength, Sanctity Laser) on one of the sides. The stimulation side was assigned pseudorandomly across animals. Time spent in the center and peripheral part of the arena (in seconds or in % of total time spent in the arena), time spent on each side (in seconds or in % of total time spent in the arena), side crosses (number of crosses), velocity (cm/s) and different behavioral components (rearing, grooming, freezing, exploration) were measured.

## Presentation of mild foot shocks and fox odor
Mild foot shocks (-0.5–1 mA, 200 ms, 10 times, @1 min interval) were delivered 20 times with 1-min interstimulus intervals, using a Supertech Ltd. (Pécs, Hungary) BioStim STI-4a computer interface connected to a STE-8a stimulus isolator end-stage. To evoke innate fear, a threat-relevant odor 2-methyl-2-thiazoline (10 μl of 2MT, Santa Cruz Biotechnology, Inc., Dallas, US), a synthetic derivative of a fox anogenital product was delivered in front of the animal on a filter paper after a baseline recording period.

## Tetrode recording
Extracellular recordings were performed with custom-built microdrives consisting of 8 movable nichrome tetrode electrodes (diameter, 12.7 μm; Sandvik) and a 50 μm core optical fiber (Thorlabs)[25,88]. Microdrive screws were specifically designed to precisely control the descent of the tetrodes and the optic fiber into the brain; the pitch of the threading was 160 μm, therefore one eighths of a turn with the screw corresponded to 20 μm descent (M0.6 stainless steel flat head screw, 12 mm length; Easterntec, Shanghai, China). Before the tetrode surgery, we measured the protruding length of the electrodes on each microdrive with a micro-ruler (Olympus SZ61 stereomicroscope; micro-ruler, Electron Microscopy Tools). The electrodes were gold-plated (PEG was dissolved in DI water to create 1 g/l concentration, then 1.125 ml of PEG-solution was mixed with 0.375 ml gold plating solution, Neuralynx, see also Ferguson et al., 2009[94] to impedances of 30-100 kΩ measured at 1 kHz (NanoZ) and dipped in DiI (ThermoFischer Scientific) red fluorescent dye to aid later track reconstruction efforts.

Before each recording session, the microdrive was connected (Omnetics) to a 32-channel RHD headstage (Intan). Data were digitized at 30 kHz and transferred from the headstage to a data acquisition board (Open Ephys) via a Serial Peripheral Interface cable (Intan). The tetrodes were advanced 0–100 μm after each recording session. The descent of the tetrodes was logged each recording day. Throughout the experiments, detailed notes of the assumed brain coordinates during each recording session were taken based on the length of the tetrodes measured before the surgery, stereotaxic information from the surgery and controlled screw turns on the microdrive.

## Optogenetic tagging
The optic fiber of the microdrive (see above) ended in an FC connector (Precision Fiber Products). The optic fiber was connected with an FC-APC patch chord during recording. Before the training session began, to optogenetically tag BFPVNs, 1 ms laser pulses were applied (473 nm, Sanctity) at 20 Hz for 2 seconds, followed by 3 seconds pause, repeated 20-30 times. Light-evoked spikes and incidental artifacts were monitored online using the OPETH plugin (SCR_018022)[95] for Open Ephys and when light-induced photoelectric artifacts or population spikes were observed, laser power was adjusted as necessary to avoid masking of individual action potentials. The significance of photoactivation was assessed with offline data analysis by applying the SALT test, which compares spike latency distributions after light pulses to a surrogate distribution based on Jensen-Shannon divergence (information radius)[88,96]. Neurons with $p < 0.01$ were considered light-activated, and thus BFPVNs. BFPVNs recorded on the same tetrode within 200 μm dorso-ventral distance were compared by waveform correlation and autocorrelogram similarity[60,97], and similar units were counted towards the sample size only once.

## Fiber photometry
A dual fiber photometry setup (Doric Neuroscience) was used to perform bilateral fluorescent calcium measurements. Recording sessions were visualized using Doric Studio Software. Two LED light sources (465 nm, 405 nm) were amplitude-modulated by the command voltage of the two-channel LED driver (LEDD_2, Doric Neuroscience, the 465 nm wavelength light was modulated at 208 Hz and 405 nm wavelength was modulated at 572 Hz) and were channeled into fluorescent Mini Cubes (iFMC4, Doric Neuroscience). Light was delivered into the HDB, MS, CA1 or retrosplenial cortex via 400 μm core patch cord fibers that were connected to optical fiber implants during training sessions. The same optical fibers were used to collect the emitted fluorescence signal from the target areas. The emitted fluorescent signal was detected with 500–550 nm fluorescent detectors integrated into the Mini Cubes. The sampling rate of emitted signals was set to 12 kHz, decoded in silico and saved in a *.csv format.

## Perfusion
Mice were anesthetized with 2% isoflurane followed by an intraperitoneal injection of a mixture of ketamine-xylazine and promethazinium-chloride (83 mg/kg, 17 mg/kg and 8 mg/kg, respectively). After achieving deep anesthesia, mice were perfused transcardially (by placing the cannula into the ascending part of the aorta via an incision placed on the left ventricle wall) with saline for 2 min, followed by 4% paraformaldehyde (PFA) solution for 40 min, then saline for 10 min. After perfusion, mice were decapitated, and brains were carefully removed from the skull and postfixed in PFA overnight. Brains were then washed in PBS and cut into 50 μm thick coronal sections using a vibrating microtome (Leica VT1200S). Sections were thoroughly rinsed in 0.1 M PB (3 × 10 min) and used for further immunohistochemical experiments.

## Acute slice electrophysiology
Animals. PV-IRES-Cre animals ($n = 7$ males, P30-60 days) were injected into the HDB (antero-posterior 0.75 mm, lateral 0.60 mm; dorsoventral 5.00 mm) bilaterally with 70 nl AAV5-EF1a-DIO-ChR2-mCherry (UNC Vector Core, US) or ssAAV-9/2-hEF1α-dlox-hChR2(H134R)-mCherry-dlox-WPRE-hGHp(A) (VVF-Zürich, Switzerland) to enable channelrhodopsin-assisted circuit mapping. After 3–6 weeks of expression time, acute brain slices were prepared for electrophysiology experiments.

Slice preparation. Mice were decapitated under deep isoflurane anesthesia. The brain was removed and placed into an ice-cold cutting solution, which had been carbogenated (95% $O_2$–5% $CO_2$) for at least 30 min before use. The cutting solution contained the following (in mM): 205 sucrose, 2.5 KCl, 26 NaHCO3, 0.5 CaCl2, 5 MgCl2, 1.25 NaH2PO4, 10 glucose. Coronal slices of 300 μm thickness were cut using a Vibratome (Leica VT1200S). After acute slice preparation, slices were placed into an interface-type holding chamber for at least 1 h of recovery time. This chamber contained standard ACSF solution at 35 °C which gradually cooled down to room temperature. The ACSF solution contained the following (in mM): 126 NaCl, 2.5 KCl, 26 NaHCO₃, 2 CaCl₂, 2 MgCl₂, 1.25 NaH₂PO₄, 10 glucose, saturated with carbogen gas as above.

In vitro electrophysiology recordings. Recordings were performed under visual guidance using Nikon Eclipse FN1 microscope with infrared differential interference contrast (DIC) optics. The flow rate of the ACSF was 4–5 ml/min at 30–32 °C (Supertech Instruments, Pecs, Hungary). Patch pipettes were pulled from borosilicate capillaries (with inner filament, thin-walled, outer diameter (OD) 1.5) with a PC-10 puller (Narishige, Tokyo, Japan). Pipette resistances were 3–6 MΩ when filled with intrapipette solution. The composition of the intracellular pipette solution was as follows (in mM): 54 d-gluconic acid potassium salt, 4 NaCl, 56 KCl, 20 Hepes, 0.1 EGTA, 10 phosphocreatine di(tris) salt, 2 ATP magnesium salt and 0.3 GTP sodium salt; with 0.2 % biocytin; adjusted to pH 7.3 using KOH and with osmolarity of ~295 mOsm/l. Recordings were performed with a Multiclamp 700B amplifier (Molecular Devices, San Jose, US), digitized at 20 kHz with Digidata analog-digital interface (Molecular Devices), and recorded with pClamp11 Software suite (Molecular Devices). For slice illumination, we used a blue LED light source (Prizmatix Ltd., Holon, Israel) integrated into the optical light path of the microscope. 2 ms long light pulses were delivered @ 5-10-20-40 Hz. To record optogenetically evoked IPSCs, cells were clamped to −65 mV, in the presence of fast excitatory synaptic transmission blockers 2,3-dihydroxy-6-nitro-7-sulfamoyl-benzo(f)quinoxaline-2,3-dione (NBQX, 20 μM) and 2-amino-5-phosphonopentanoic acid (AP-5, 50 μM).

Immunohistochemical identification of in vitro recorded cells. After acute slice electrophysiology experiments, brain sections were fixed overnight in 4% PFA. Sections were extensively washed in 0.1 M PB and TBS and blocked in 1% human serum albumin (HSA; Sigma-Aldrich) solution for 1 h. Then, sections were incubated in primary antibodies against PV, SOM, CR or ChAT (see Table S2) for 48–60 h. This step was followed by thorough rinse with TBS (3 × 10 min) and overnight incubation with a mixture of secondary antibodies (see Table S3) and streptavidin-A488 (Invitrogen, 1:1000). We used 0.1% Triton-X detergent through every incubation step due to the thickness of the brain section. Finally, sections were washed in TBS and PB, mounted on microscopy slides and covered with Vectashield (Vector Laboratories Inc, US) and imaged with a Nikon A1R confocal laser scanning microscope.

## Histological track reconstruction

After tetrode recording and fiber photometry experiments, animals were anesthetized with an intraperitoneal injection of ketamine-xylazine (83 mg/kg and 17 mg/kg, respectively) and underwent an electrolytic lesioning protocol (30 μA for 5 s on two leads of two selected tetrodes, where opto-tagged BFPVNs were recorded; stimulator from Supertech, Pecs, Hungary). Twenty-to-thirty minutes after the electrolytic lesion, mice were perfused transcardially as described above. Microdrives and/or optical fibers were extracted and the brain was gently removed from the skull. The brain was postfixed in 4% PFA overnight and then washed in 0.1 M phosphate buffer (3 × 10 min). The explanted microdrives were examined under stereomicroscope and the protruding length of the electrodes were measured and compared to the depth registrations made during the recording sessions.

Coronal sections of 50 μm thickness were cut by a vibratome (Leica VT1200S). Special care was taken to section the brain perpendicular to brain surface, so that resulting sections could be easily aligned with coronal atlas images. The sections were thoroughly washed in phosphate buffer 3 times (10 min each) and mounted on microscopy slides in Aquamount mounting resin. Fluorescent micrographs of the sections were taken using a Nikon C2 confocal microscope. Four times four large field-of-view dark-field images, red and green fluorescent images were taken at 10x magnification in order to capture the whole sections.

Confocal microscopic images were further analyzed to reconstruct in vivo recording locations. The recording locations were referenced to atlas coordinates[98]. By referencing recording location to atlas coordinates, individual size differences of mouse brains compared to the atlas reference and slight deviations from the vertical direction during electrode descent were accounted for. Dark-field whole-section brain images aided atlas alignment, since they provided a detailed contrast for white and gray matter. By using Euclidean transformations, atlas images of coronal sections were morphed on the corresponding dark-field brain images, to verify area boundaries and determine the coronal plane of the section. If the brain section was non-uniformly distorted during histological processing, special care was taken to accurately map the vicinity of the electrode tracks within the target areas. Next, green fluorescent images of the same sections were transformed similarly as the dark-field images to verify PV expression within the boundaries of anatomical structures corresponding to the atlas images. In the case of tetrode recordings, aligned atlas images were superimposed on red fluorescent images of the same field of view, which showed the DiI-labeled electrode tracks. Coordinates of electrode entry points and deepest points in the brain were marked by DiI tracks and the tip of the electrode was localized using small electrolytic lesions. Coordinates of electrode endpoints were read from the atlas image and were used to interpolate the recording locations referenced to the atlas coordinates, based on experimental logs of the protrusion of the electrode. Based on this interpolation procedure, antero-posterior, lateral and dorso-ventral coordinates and an atlas-defined brain area were assigned to each recording session and thus to each recorded neuron.

Sections from mice implanted with optic fibers were processed similarly; dark-field whole section brain images were used for atlas alignment, while green fluorescent images verified the expression of ChR2, ArchT or GCaMP6s. Optic fiber locations were reconstructed based on the tissue track caused by the fiber.

## Anterograde tracing

Sections were washed in PB (0.1 M) and immersed in 30% sucrose solution overnight, then freeze-thawed over liquid nitrogen three times. Sections were extensively washed in 0.1 M PB and TBS and blocked in 1% human serum albumin (HSA; Sigma-Aldrich) solution for 1 h. Then, sections were incubated in primary antibodies against choline-acetyltransferase (ChAT) or parvalbumin (PV) or calretinin (CR) together with anti-EGFP antibody (see Table S2) for 48–60 h. Sections were rinsed 3 times for 10 min in TBS; secondary fluorescent antibodies (see Table S3) were applied overnight. Sections were rinsed in TBS and 0.1 M PB and mounted on slides in Aquamount mounting medium (BDH Chemicals Ltd). Next, fluorescent images were taken with a Nikon A1R Confocal Laser Scanning Microscope. In target areas of BFPVNs, fluorescent images of 10x magnification were taken. The axon densities of BFPVN projections were estimated by the mean pixel brightness value in each region.

## Retrograde tracing

Sections were washed in 0.1 M PB, incubated in 30% sucrose overnight for cryoprotection, then freeze-thawed over liquid nitrogen three times for antigen retrieval. Sections were subsequently washed in PB and Tris-buffered saline (TBS) and blocked in 1% HSA in TBS, then

incubated in a mixture of primary antibodies for 48–72 h. This was followed by extensive washes in TBS, and incubation in the mixture of appropriate secondary antibodies overnight. Then, sections were washed in TBS and PB, dried on slides and covered with Aquamount.

The combinations and specifications of the primary and secondary antibodies used are listed in Tables S2 and S3. The specificities of the primary antibodies were extensively tested, using knock-out mice if possible. Secondary antibodies were tested for possible cross-reactivity with the other antibodies used, and possible background labeling without primary antibodies was excluded.

Sections were imaged using a Nikon Ni-E Eclipse epifluorescent microscope system equipped with a 10× air objective and a Nikon DS-Fi3 camera. Fifty μm thick coronal sections spaced at 300 μm were prepared, so that approximately one-sixth of the whole brain was sampled to measure and estimate the number of transsynaptically labeled input cells.

### Correlated light an electron microscopic analysis

Sections from the anterograde tracing experiments (see above) were imaged with Nikon A1R confocal laser scanning microscope for axon density measurements and for imaging target cell types. First, sections from anterograde tracing experiments were used for immunohistochemical staining against markers of possible target cell types (PV; VIP, vasoactive intestinal peptide; CB, calbindin; CR, calretinin; ChAT). A subset of sections were mounted in Aquamount mounting medium and imaged at 60× magnification and oil immersion with A1R Nikon Confocal laser scanning microscope. Sections from at least 3 animals were examined for possible contact sites between BFPVN projecting axons (labeled by eGFP) and target cell types. When possible terminal contacts were observed with a target cell type, another subset of sections from the same immunohistochemical experiment were mounted on slides in 0.1 M PB and were imaged similarly as described above. Possible terminal contacts were photographed at multiple magnifications (60×, 20×, 10×, and 4×) and surrounding landmarks blood vessels, location of ventricles, corpus callosum and other white matter tracts, tears at the edge of the section) were marked. After imaging, the sections were demounted from the slides to perform a second immunohistochemical staining to label eGFP (labeling BFPVNs and their projections) with 3,3′-Diaminobenzidine (DAB) as chromogen. Sections were rinsed extensively in TBS (3 × 10 min), then anti-chicken biotinylated secondary antibody was applied (in order to label eGFP-containing fibers, which were previously recognized by a chicken anti-eGFP primary antibody, see above) overnight. This step was followed by incubation in avidin–biotinylated horseradish peroxidase complex overnight (Elite ABC; 1:500; Vector Laboratories); the immunoperoxidase reaction was developed using DAB (Sigma-Aldrich) chromogen. Next, sections were mounted on slides in PB and light microscopic imaging (Nikon C2) was performed to reveal BFPVN axons. Next, light microscopic images were taken and aligned with former fluorescent images based on previously marked landmarks to determine the precise location of possible contact sites between BFPVN axons and target cell types on the light microscopic images. After this, sections were incubated with 0.5% osmium tetroxide in 0.1 M PB on ice and they were dehydrated using ascending alcohol series and acetonitrile, and finally incubated in 1% uranylacetate in 70% ethanol and embedded in Durcupan resin (ACM; Fluka). Brain regions containing the possible contact sites were re-embedded into resin blocks and 70 nm thick sections were cut with an ultra-microtome (Leica EM UC 6). Electron microscopic images were taken either with a Hitachi H-7100 transmission electron microscope or a FEI series electron microscope to reveal synaptic contacts between BFPVNs and target cell types in the MS, CA1 and retrosplenial cortex.

### Data analysis

**Analysis of extracellular recordings.** Data analysis was carried out using custom written Matlab code (R2016a, Mathworks). Tetrode recording channels were digitally referenced to a common average reference, filtered between 700 and 7000 Hz with Butterworth zero-phase filter and spikes were detected using a 750 μs censoring period. Action potentials were sorted into putative single neurons manually by using MClust 3.5 (A.D Redish, University of Minnesota). Autocorrelations were examined for refractory period violations and only putative units with sufficient refractory period were included in the data set. Neurons with $p < 0.01$ significance values of the SALT statistical test (H-index) were considered light-activated, therefore BFPVNs (see above). Spike shape correlations between light-induced and spontaneous spikes were calculated for all BFPVNs. The correlation coefficient exceeded R = 0.84 in all and R = 0.9 in 22/36 optotagged neurons (0.96 ± 0.0085, median ± SE; range, 0.8427–0.9989, see also Cohen et al.[99]). Only those BFPVNs with high cluster quality (isolation distance > 20 and L-ratio <0.15 calculated based on full spike amplitude and first principal component of the waveform) were included in the final dataset for further analysis[25,100].

After spike sorting, the activity of individual neurons was aligned to cue presentation, reward and punishment delivery. Statistics were carried out on each neuronal unit; baseline activity was defined by taking a 1 s window before the cue, then firing rate in the baseline window was compared to firing rate in the test window (0–0.5 s after the event). One-sided hypotheses of firing rate increase and decrease were tested by Mann–Whitney $U$ test ($p < 0.001$; for cue-evoked activity, separately for likely reward and likely punishment cue). Neurons were sorted into different groups based on their statistically significant responses to the behaviorally relevant events (e.g. activated by cue, inhibited by reward, etc.).

We calculated event-aligned raster plots and peri-event time histograms of spike times (PETHs) for all neurons, and lick times for all recording sessions, aligned to both behavioral events and/or photostimulation pulses. PETHs were smoothed with a Gaussian kernel. To calculate average PETHs, individual PETHs were Z-scored by the mean and standard deviation of a baseline window (1 s before cue onset) and averaged across neurons. Response latency and jitter to optogenetic stimulation and behavioral events were determined based on activation peaks in the PETHs. We performed K-means clustering of Z-scored PETHs (similar to ref. 99) aligned to the auditory cues for n = 685 well-isolated neurons recorded from the HDB of task-performing mice. We defined five separate clusters based on the first 3 principal components of the PETHs triggered on the two auditory cues. We ordered the clusters according to the percentage of tagged BFPVNs they contained and sorted the neurons within the clusters according to the principal components.

Autocorrelograms (ACG) were calculated at a 0.5 ms resolution. Burst index (BI) was calculated by the normalized difference between maximum ACG for 0-10 ms lags and mean ACG for 180–200 ms lags, where the normalizing factor was the greater of the two numbers, yielding an index between −1 and 1[101]. A neuron with a BI > 0.3 and a refractory period <2 ms was considered bursting, confirmed by the presence of burst shoulders on average ACG in the bursting group and the complete lack of burst shoulders on the average ACG in the non-bursting group. To examine burst coding in BFPVNs, analysis of neuronal responses to punishment were also carried out when only burst spikes or single spikes were considered for a neuron. A burst was detected whenever an inter-spike interval (ISI) was <10 ms and subsequent spikes were considered as part of the burst as long as the ISI remained <15 ms. Data were Z-score normalized with their surrogate mean and standard deviation to plot average ACG. The surrogates were generated according to ref. 102.

**Fiber photometry signal processing.** Fiber photometry signals were preprocessed according to refs. 103. Briefly, the recorded fluorescence signals were filtered using a low-pass Butterworth digital filter below 20 Hz to remove high-frequency noise. To calculate dff, a least-squares linear fit was applied to the isosbestic 405 nm signal to align its

baseline intensity to that of the calcium-dependent 465 nm ($f_{465}$) signal. The fitted 405 nm signal ($f_{405,\text{fitted}}$) was used to normalize the 465 nm signal as follows:

$$\text{dff} = \frac{f_{465} - f_{405,\text{fitted}}}{f_{405,\text{fitted}}} * 100 \tag{1}$$

to remove the effect of motion and autofluorescence. A 0.2 Hz high pass Butterworth digital filter was used to filter out the slow decrease of the baseline activity observed during the recording session. Finally, the dff signal was aligned to cue and reinforcement times, smoothed with a Gaussian kernel (width, 100 ms), Z-scored by the mean and standard deviation of a baseline window (1 s before cue onset) and averaged across sessions.

**Statistics.** Firing rates and other variables were compared across conditions using non-parametric tests (normality of the underlying distributions could not be determined unequivocally). For non-paired samples, Mann–Whitney $U$ test was used; for paired samples, Wilcoxon signed-rank test was applied (two-sided unless stated otherwise). Pearson's correlation coefficient was used to estimate correlations between variables, and a standard linear regression approach was used to judge their significance (one-sided F-test, in accordance with the asymmetric null hypothesis of linear regression). Peri-event time histograms (PETHs) show mean ± SE. Box-whisker plots show median, interquartile range and non-outlier range, with all data points overlaid. Bar graphs show mean and all data points overlaid.

ROC analysis was performed to test if the anticipatory lick rate difference between Cue 1 and Cue 2 trials was different between control and ArchT groups in the optogenetic inhibition experiment. First, average PETH of lick rates were calculated for control and ArchT groups in a 1.1 s long time window from cue onset (from cue to reinforcement), then values at every 10 ms of the time window were taken. Next, area under the ROC curve (auROC) based on the cumulative density functions of lick rate differences in control and ArchT groups were calculated for each time point. The significance of the auROC was tested with a bootstrap permutation test with 200 resamplings.

### Reporting summary
Further information on research design is available in the Nature Portfolio Reporting Summary linked to this article.

## Data availability
The in vivo electrophysiology, fiber photometry and behavioral data generated in this study have been deposited at https://doi.org/10.6084/m9.figshare.22776776.v1, https://doi.org/10.6084/m9.figshare.25403350 and https://doi.org/10.6084/m9.figshare.25403593. Source data are provided with this paper.

## Code availability
MATLAB codes generated for this study are available at https://github.com/hangyabalazs/PV_Pavlovian_analysis and also https://doi.org/10.5281/zenodo.10808341.

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

## Acknowledgements

The authors thank Katalin Lengyel for technical assistance in anatomical methods, Ilka Jakab and Melinda Szabó for assistance in behavioral training and Dr. Norbert Hájos for helpful comments on the manuscript. We thank the FENS-Kavli Network of Excellence for fruitful discussions. This work was supported by the "Lendület" Program (LP2015-2/2015), NKFIH K135561, NKFIH K147097 and the European Research Council Starting Grant no. 715043 to B.H.; the Hungarian Brain Research Program

NAP3.0 (NAP2022-I-1/2022) by the Hungarian Academy of Sciences and the European Union project RRF-2.3.1-21-2022-00004 within the framework of the Artificial Intelligence National Laboratory to B.H. and G.N.; Frontline Research Excellence Programme by the Hungarian National Research, Development and Innovation Office (NRDI Fund 133837) and the European Union project RRF-2.3.1-21-2022-00011 within the framework of the Translational Neuroscience National Laboratory to GN; ÚNKP-21-3 New National Excellence Program of the Ministry for Innovation and Technology and the Kerpel Scholarship of Semmelweis University to P.H; ÚNKP-23-4-II New National Excellence Program of the Ministry for Culture and Innovation from the source of the National Research, Development and Innovation Fund to D.S. We acknowledge the help of the Nikon Center of Excellence at the HUN-REN Institute of Experimental Medicine (IEM), Nikon Europe, Nikon Austria, and Auro-Science Consulting for kindly providing microscopy support and the supportive help of the Central Virus Laboratory and the Behavioral Unit of IEM. We thank István Katona for kindly providing access to an in vitro electrophysiology setup. We thank Mackenzie Mathis, Ann Kennedy and Ethan Tyler for open access science art at SciDraw.

## Author contributions

P.H. and B.H. conceived the study; P.H. performed the immunohistological experiments, fluorescent and electron microscopic imaging; M.M and G.N. performed the retrograde tracing experiments; P.H., I.S., A.V., and Z.Z. performed the recording and optogenetic activation experiments; P.H., A.V. and V.L. conducted the fiber photometry and optogenetic inhibition experiments; D.S. conducted the in vitro electrophysiology experiments; P.H., B.K., D.S. and V.L. analyzed the data; P.H., B.K., D.S., M.M., and G.N. prepared the figures; B.H. and P.H. wrote the manuscript with inputs from all authors.

## Competing interests

The authors declare no competing interests.

## Additional information

**Peer review information** : *Nature Communications* thanks Shih-Chieh Lin and the other, anonymous, reviewer(s) for their contribution to the peer review of this work. A peer review file is available.

