## [Peer Review File · Nature Communications]

Parvalbumin-expressing basal forebrain neurons mediate learning from negative experienceReviewers' Comments:

Reviewer #1:

Remarks to the Author:

Hegedüs and colleagues reported that Parvalbumin-expressing basal forebrain neurons (BFPVNs) mediate learning from negative experiences. The authors found that BFPVNs preferentially showed phasic activation to punishment, but only slow and delayed responses to reward, and optogenetic inhibition of BFPVNs during punishment impaired the formation of cue-outcome associations. The authors also mapped the input-output connectivity of BFPVNs by anterograde and mono-transsynaptic retrograde tracing experiments. Finally, it is suggested that the arousing effect of BFPVNs is recruited by aversive stimuli to serve crucial associative learning functions during awake behaviors.

Overall, I would say that the current work has provided some useful information. However, a few technical and conceptual issues should be addressed.

Major comments:

1. Why choose PV neurons but not other cell types in the BF? I understand that the role of cholinergic neurons has been examined using the same task in previous work. However, cholinergic neurons have a much wider interest in the field. I am not convinced that characterizing the BFPVNs has a major significance.
2. Is the observed response unique to PV neurons? How about other cell types?
3. Is the observed response in PV neurons a learned behavior?
4. Mapping of the input-output of BFPVNs has been performed in previous work. I wonder about the necessity of repeating these experiments.
5. The authors used fiber photometry recording to measure the calcium signal from axons of BFPVNs, and found a broadcasting pattern. Further manipulation experiments should be performed to identify the downstream brain regions that mediate the behavior effect.
6. The authors used EM to identify synaptic contact of BFPVNs. However, these experiments did not provide too much information. They should perform a more systematic analysis of the postsynaptic target neurons. I think EM may not be the right method to use for this purpose, and an alternative is to perform brain slice electrophysiology recording.
7. For the fiber photometry recording experiments, the raw signal trace should be provided rather than only showing an averaged plot.
8. The author found that BFPVNs receive inputs from aversion-coding nuclei. How do the authors know these neurons actually provide the aversive information to BFPVNs?
9. Further details of the optogenetic inhibition experiments should be provided. At least, a daily learning curve from the two groups during learning should be provided
10. how do the authors explain that BFPVNs broadcast aversive information, but activating them did not cause aversive behavior?

Specific comments:

In Fig. 3, the cartoon in panel A provides little information. Also, a diagram to indicate the timing of laser stimulation should be added.

Fiber photometry recording of calcium signal is not 'imaging'. This should be corrected.

In some experiments, the number of mice equals 2?

Reviewer #2:

Remarks to the Author:

Hegedüs et al investigated how basal forebrain parvalbumin neurons (BFPVNs) play a role in associative learning. Previous studies suggest that BFPVNs regulate cortical gamma oscillation and promote wakefulness. However, how these neurons participate in awake behavior remains unclear. To answer this question, the authors first recorded the activity of phototagged BFPVNs in HDB while head-fixed mice performed an auditory probabilistic Pavlovian conditioning task, in which two auditory cues were associated with different outcome probabilities (80% reward / 10% airpuff / 10% omission for cue1 and 25% reward / 65% airpuff / 10% omission for cue2). BFPVNs showed phasic activation to airpuff, and such responses were not modulated by expectation. Optogenetic inhibition of BFPVNs during punishment impairs behavioral discrimination of the two cues. However, optogenetic activation of BFPVNs did not generate place aversion. The authors further mapped the inputs and outputs of these neurons, and show that BFPVNs receive inputs from aversion-related nuclei, such as medial raphe. GCamp6 photometry recording confirmed that BFPVNs were excited by airpuff, and broadcast this signal to its main downstream targets, including medial septum, hippocampus and retrosplenial cortex. Based on these results, the authors concluded that BFPVNs broadcast aversive information to downstream targets to promote arousal and mediate learning from negative experience. While phototagging of BFPVNs in behaving animals provides valuable information about how these neurons operate under behavioral contexts, there are considerable inconsistencies in the results that made me question their interpretation. Here are my comments:

Major comments:

1. A key result of the study is that optogenetic inhibition of BFPVNs during airpuff impairs behavioral discrimination of the two cues. This manipulation eliminated differences in anticipatory licking patterns and reaction time (Figure 3). However, even if ArchT inhibition during airpuff presentation abolished the effect of the negative outcome, the two cues were still different in terms reward probabilities (80% vs 25%). Shouldn't the differential reward alone be sufficient to endow different behavioral response patterns to the two cues? The lack of behavioral difference in Fig 3C cannot be explained by the lack of punishment effect alone. Instead, it would be more compatible with a failure to discriminate between the two cues (sounds).
2. The lack of effect on real-time place preference/aversion (Fig S4) further support that BFPVNs neither encode aversive nor rewarding information. Are they any example of aversive-encoding neurons whose activation does not lead to behavioral aversion?
3. Is it possible that BFPVNs were responding to sound intensity instead of airpuff? Airpuff delivery should induce a much louder sound than either cue1 or cue2. Such auditory responses were reported in basal forebrain cholinergic neurons and might not be unexpected to observe in BFPVNs (Guo W, Robert B, Polley DB. The Cholinergic Basal Forebrain Links Auditory Stimuli with Delayed Reinforcement to Support Learning. *Neuron*. 2019. doi:10.1016/j.neuron.2019.06.024). Could BFPVN manipulations exert their effect through modulating BF cholinergic neurons?
4. Several previous studies have emphasized the role of BFPVNs in arousal through their projections to the cerebral cortex (Kim et al, *PNAS*, 2015; Xu et al, *Nat Neurosci* 2015; McKenna et al 2020). But the current results only find frontal cortex to be a minor target of BFPVNs (line 274), and there is little effect that BFPVNs would lead to increased arousal. In Fig S4 (RTPP), there was also no effect on movement pattern. How do the authors interpret this discrepancy?
5. PV neurons are ubiquitously present in all brain circuits. The location of BFPVNs should be carefully characterized in its various experiments (phototagging, optogenetics, GCamp6, viral tracing) to make sure that the results point to the same population of PVNs. The use of 300nl GCamp6 virus raises the concerns of a large infection volume beyond the HDB region. Do the author expect some of the results may result from heterogeneity of the underlying neuronal population?

Other comments:

1. What's distinction between no-lick period and foreperiod in the task design (Fig 1a)? They seem redundant.
2. The difference in licking pattern at the population level (Fig 1d) was much smaller than the example session in Fig 1c. The authors might want to consider showing a more representative session in Fig 1c.

3. Figure 1f and its inset do not match up. Which area was magnified?
4. Phototagging provides highly valuable basic physiological characterizations of PV neurons in the BF. The authors should provide a table showing the raw firing rate characteristics of all phototagged neurons (baseline firing rates, ISI distribution, peak firing rate....), in addition to the normalized firing rate in Fig 1j.
5. If BFPVN responses to airpuff were not modulated by surprise (Fig S2), what could explain the difference in response amplitude and baseline firing rates between the first and second half of the task (Fig 2g-i)?
6. The differences in anticipatory licking in Fig 3b seemed to arise immediately after cue onset in the control group. Yet it only reached statistical significance much later (0.6-1.1s), and the RTs (Fig 3d) were also much slower in both groups compared to the population average in Fig 3b, 3c. This discrepancy should be addressed.
7. The detailed of the optogenetic inhibition should be provided: was inhibition delivered throughout all stages of the training? Or just the last stage? Was the behavioral difference present during the first session of the last stage? Were there any behavioral difference at earlier stages of training?
8. Vglut3 immunostaining in Fig 4f: How does Vglut3 staining help determine whether the neuron expresses Vglut3? I thought Vglut3 should be present mostly in the synapse and not in the soma, just like Vglut2?
9. The spread of GCamp6 viral expression should be further characterized to minimize the effect of injecting PV neurons in nearby brain regions.
10. Fig 5d: vGlut1+ neuron image missing.
11. GCamp6 responses sometimes start before time zero (Fig 6b, S6). Where the traces smoothed? By what kind of smoothing kernel?

Rebuttal letter: Parvalbumin-expressing basal forebrain neurons mediate learning from negative experience

Structure:

Reviewer comments: black, italic

Our responses: blue

We thank the Reviewers for their constructive comments and the Editor for the opportunity to address the Reviewer's points in this rebuttal. We performed additional experiments, data analyses and refined the interpretation of our results.

1. We performed bulk calcium recordings from somatostatin-expressing GABAergic neurons of the BF (BFSOMNs) while mice performed the probabilistic Pavlovian conditioning task. We found that BFSOMNs differed from BFPVNs in their relative response magnitudes (e.g. large reward responses for BFSOMNs) and coding properties (differential response to surprising and expected outcomes in BFSOMNs).
2. We clustered the spiking patterns of electrophysiologically recorded non-identified BF neurons, which revealed multiple clusters of BF neurons with distinct response properties.
3. We performed *in vitro* slice electrophysiology experiments in the major target regions of BFPVNs including the CA1 area of the hippocampus, medial septum and the retrosplenial cortex that revealed differential downstream impact of BFPVN projections.
4. We performed calcium measurements of BFPVNs after electrical shock and in the presence of predator odor, revealing that BFPVNs respond to aversive stimuli in different sensory modalities.
5. We performed additional analyses and refined our interpretations in response to both Reviewers' comments.

Please see our point-by-point responses below.

Responses to comments from Reviewer #1

Hegediis and colleagues reported that Parvalbumin-expressing basal forebrain neurons (BFPVNs) mediate learning from negative experiences. The authors found that BFPVNs preferentially showed phasic activation to punishment, but only slow and delayed responses to reward, and optogenetic inhibition of BFPVNs during punishment impaired the formation of cue-outcome associations. The authors also mapped the input-output connectivity of BFPVNs by anterograde and mono-transsynaptic retrograde tracing experiments. Finally, it is suggested that

the arousing effect of BFPVNs is recruited by aversive stimuli to serve crucial associative learning functions during awake behaviors.

Overall, I would say that the current work has provided some useful information. However, a few technical and conceptual issues should be addressed.

We thank the Reviewer for their insightful, constructive comments based on which we could significantly improve our manuscript, and for finding our work useful.

Major comments:

1. Why choose PV neurons but not other cell types in the BF? I understand that the role of cholinergic neurons has been examined using the same task in previous work. However, cholinergic neurons have a much wider interest in the field. I am not convinced that characterizing the BFPVNs has a major significance.

We appreciate the Reviewer's concern that BFCNs may have a wider interest in the field than BFPVNs. Historically, cholinergic neurons have been in the limelight as acetylcholine was the first molecule isolated (Sir Henry Dale, 1914) and identified as a neurotransmitter (in the periphery, Otto Loewi, 1921), and later confirmed as a central neurotransmitter by the first efficient anti-ChAT antibodies (McGeer et al., 1974). However, since Barbara Jones (Gritti et al., 2006) and others (Yang et al., 2017) pointed out that cholinergic neurons are outnumbered by GABAergic and glutamatergic neurons in the basal forebrain and lesioning BF GABAergic cells leads to cognitive deficits (Yoder and Pang, 2005; Dwyer et al., 2007; Roland et al., 2014), there has been a surge of interest to study these previously overlooked cell types. In 1988, Freund and Antal showed that the parvalbumin (PV)-expressing subpopulation of BF GABAergic neurons innervate hippocampal interneurons, which ignited a long line of research of medial septal (anterior BF) PV neurons in hippocampal rhythmogenesis (Freund and Antal, 1988; Serafin et al., 1996; Bland et al., 1999; Hangya et al., 2009; Joshi et al., 2017). As the focus gradually shifted towards the neocortex from the hippocampus, a parallel line of research established the role of BFPVNs in controlling cortical gamma oscillations (Kim et al., 2015; Yang et al., 2017; Maness et al., 2022). A number of studies examined the local connectivity and firing properties of BFPVNs and revealed their roles in sleep regulation (Xu et al., 2015; Yang et al., 2017), as well as suggested they might have important roles in attention (McKenna et al., 2013; Kim et al., 2015; Yang et al., 2017; Lozano-Montes et al., 2020) and learning (Dwyer et al., 2007; Lin and Nicolelis, 2008; Avila and Lin, 2014; Roland et al., 2014).

Finally, BFPVNs decrease in number through aging, thus having important medical relevance (Stanley et al., 2012; Bañuelos et al., 2023; Chaves-Coira et al., 2023). Moreover, the recently discovered possibility that oscillatory control may provide disease-modifying therapies for Alzheimer's Disease (Iaccarino et al., 2016; Martorell et al., 2019), for which controlling BFPVNs could be a viable option (Etter et al., 2019), has lent special importance to this cell group. Therefore, BFPVNs, a numerous cell group with specific cortical and subcortical connectivity and

roles in cognitive function, are receiving rapidly increasing attention. Under these circumstances, demonstrating that they exhibit specific firing patterns essential for associative learning is expected to be of wide interest. We edited the Discussion and Introduction sections of our manuscript to better reflect these thoughts.

2. Is the observed response unique to PV neurons? How about other cell types?

We thank the Reviewer for this question, which we found very exciting, and explored at depth. First, we performed an unbiased analysis of all well-isolated neurons ($n = 685$) recorded during the probabilistic Pavlovian conditioning task. Second, we performed bulk calcium recordings of another prominent GABAergic cell type of the basal forebrain, the somatostatin-expressing (SOM) neurons (Yang et al., 2017). We present our novel findings below and also in the revised manuscript (Fig.S3 and S4, Results, Methods, Discussion). These new experiments increase the breadth of our study, expected to engage a broad audience interested in circuit mechanisms of cognition.

We performed K-means clustering of Z-scored PETHs (similar to (Cohen et al., 2012)) aligned to the auditory cues for $n = 685$ neurons recorded from the HDB of task-performing mice. We defined five separate clusters based on the first 3 principal components of the PETHs triggered on the two auditory cues. Optogenetically identified BFPVNs were mostly distributed in cluster #1 and #2 (12 of all 36 tagged BFPVNs in a cluster of 134 HDB cells and 17 of all 36 tagged BFPVNs in a cluster of 193 HDB cells, respectively). These clusters were characterized by prominent responses to punishment and smaller responses to reward and outcome-predicting auditory cues, reproducing the response profiles of optogenetically identified BFPVNs (Fig. 2). Non-tagged neurons in clusters #1 and #2 could represent both undetected BFPVNs due to limitations of viral infection and photoactivation, and possibly other HDB neurons activated by reinforcement. Since these BFPVN-containing clusters contained 48% of all recorded BF neurons, while anatomical estimates of PV-expressing neurons are usually lower (Gritti et al., 2006; Yang et al., 2017), the latter scenario is likely. Of note, cluster #1 also contained neurons that had comparable responses to reward and punishment, arguing for the presence of other cell types in this cluster (see also below).

BF neurons of cluster #5 showed fast and precise responses to both punishment and reward (Table R1). They also responded to outcome-predicting cues with a preference towards the sound cue forecasting likely reward. We previously identified BF neurons exhibiting this distinctive time course of firing responses as BF cholinergic neurons (BFCNs, (Hangya et al., 2015; Laszlovszky et al., 2020; Hegedüs et al., 2023)). In accordance, this cluster did not contain any identified BFPVNs.

Cluster	Expected reward	Expected punishment
#1	0.193±0.161 (s)	0.085±0.085 (s)
#2	0.274±0.171 (s)	0.151±0.130 (s)
#5	0.097±0.126 (s)	0.028±0.010 (s)

Table R1. Peak latency of activation after reward and punishment.

Cluster #4 contained neurons that showed suppression of firing after punishment and to a lesser extent after reward. These responses mirrored BFPVN response profiles, suggesting that some of these neurons might be locally inhibited by the GABAergic BFPVNs. Such connections were found to be rare by circuit mapping studies (Xu et al., 2015; Yang et al., 2017), and in accordance, this group contained only 14% of HDB neurons. Finally, neurons in cluster #2 showed no strong responses to task-relevant behavioral events.

Figure R1. K-means clustering of BF neuronal responses reveals groups of neurons with distinct firing patterns. a, Color-coded, Z-scored PETHs of all neurons ($n = 685$). Red asterisks indicate tagged BFPVNs. The clusters were ordered according to percentage of tagged neurons. **b,** Top and middle, PETH of example neurons aligned to stimulus (top) and reinforcement (middle). Bottom, Average, Z-scored PETH of each cluster. Errorshades represent SEM.

The basal forebrain contains another notable GABAergic cell type that expresses somatostatin (SOM). These basal forebrain SOM-expressing neurons (BFSOMNs) inhibit BFCNs, BFPVNs

and BF glutamatergic neurons (Xu et al., 2015) via GABAergic synapses. In addition, SOM itself presynaptically inhibits glutamate and GABA release onto BFCNs (Zaborszky et al., 2012). BFSOMNs receive excitatory inputs from local glutamatergic neurons as well as BFCNs via nicotinic ACh receptors, while muscarinic receptors convey slower hyperpolarizing inputs (Yang et al., 2017). Many of these cells are sleep-active and sleep promoting, suggesting that BFSOMNs potentially suppress all wake-promoting BF cell types during non-REM sleep (Xu et al., 2015). At the same time, specific activity of BFSOMNs during behavior was not known.

To explore the responses of BFSOMNs to behaviorally relevant events during learning, we performed fiber photometry recordings of bulk calcium responses in the HDB of SOM-Cre mice injected with AAV2/9.CAG.Flex.GCAMP6s.WPRE.SV40 (Figure R2). We found that BFSOMNs responded to rewards, punishments, and reward-predicting auditory cues. This activity pattern differentiated them from BFPVNs that were most active after punishments, and was remarkably similar to BFCN calcium responses (Hegedüs et al., 2023). However, no overlap was found between SOM- and ChAT-expressing neurons in the HDB (Fig.R2c), as reported before (Yang et al., 2017). Additionally, BFSOMNs showed higher activity after surprising than after expected rewards, similar to BFCNs and unlike BFPVNs. However, this higher activity occurred on the background of higher calcium levels at times of surprising reward delivery, due to a suppression of the calcium signal following the activation induced by the reward-predicting cue (not observed for BFCNs, (Hegedüs et al., 2023)), complicating the interpretation of this difference. Higher peak activity after expected vs. surprising punishment also suggests that the abovementioned suppression plays a role in this difference.

Thus, BFSOMNs responded to the same behaviorally salient events as BFPVNs but differed in relative response magnitudes (e.g. large reward responses for BFSOMNs) and coding properties (differential response to surprising and expected outcomes in BFSOMNs). This suggests that cluster #1 in the unbiased grouping above represents a mixed GABAergic population exhibiting reward- and punishment-related activation.

We propose that BFSOMNs might provide negative feedback onto BFCNs, both inhibiting their outputs (via GABA release) and decoupling them from their inputs (via SOM release), thereby limiting the duration of their activation. Previous recording studies showing fast and precise activation of BFCNs support this notion (Hangya et al., 2015; Laszlovszky et al., 2020; Hegedüs et al., 2023), suggesting that a fast bottom-up activation of the cholinergic system might recruit negative BFSOMN feedback via nicotinic receptors (Yang et al., 2017).

We included these new results as Figure S4 and discuss them in the Results, Methods and Discussion sections.

a**b****c****d****e****f****g****h****i****j****k****l****m**
Figure R2. BFSOMNs respond to reward, punishment, and reward-predictive auditory cues. **a**, Schematic diagram of bulk calcium measurements of BFSOMNs. **b**, Fluoromicrograph of an optical fiber track (green, GCaMP6s; asterisk, tip of the optical fiber). **c**, Immunohistochemical staining of the HDB shows no overlap between BFSOMNs and BFCNs (green, SOM; magenta, ChAT; 8 out of 392 SOM cells (~ 2%) expressed ChAT, n = 5 animals). **d**, SOM-Cre mice have learned the task indicated by higher anticipatory lick rate to the reward predicting cue (n = 20 sessions recorded from n = 4 mice). **e-g**, PETHs of bulk-calcium recording of BFSOMNs aligned to cue (left), reward (middle) and punishment (right) in an example session (top, trial-by-trial data; bottom, session average). **h-j**, Color-coded PETH showing all recorded training sessions where animals have acquired the task contingencies, aligned to cue (left), reward (middle) and punishment (right). **k-m**, Average PETHs of BFSOMN response during the task, aligned to cue (left), reward (middle) and punishment (right) with bar plots representing the peak response distribution. Wilcoxon signed-rank test; **, p < 0.01; ***, p < 0.001. Errorshades represent SEM.

3. Is the observed response in PV neurons a learned behavior?

We thank the Reviewer for raising this important question. We typically observe that outcome-related responses in the BF occur already at first presentation, while responses to sensory cues predicting future outcomes emerge by learning, confirmed by other studies (Hangya et al., 2015; Guo et al., 2019; Crouse et al., 2020). Therefore, we expected that BFPVN responses to aversive stimuli can be detected independent of learning. To test this, we analyzed whether responses to air puffs occurred after the first punishment presentation. Indeed, air puff responses were present already in the first sessions in which punishments were introduced, and even when only the first punished trials were considered (Figure R3).

We also tested whether BFPVNs respond to other aversive stimuli and found that foot shock and fox odor potentially activated BFPVNs, suggesting that they convey information that generalizes over different aversive events (see our response to point #3 of Reviewer #2). These responses also occurred already at first encounter.

We added these new results to the manuscript as a new Figure 3 and text in Results, Methods, and Discussion.

Figure R3. BFPVNs respond to aversive stimuli of different modality (air puff, foot shock, fox odor) in naïve mice. **a**, Schematic diagram of introducing punishment during the Pavlovian conditioning task. **b**, Example of a session when punishment was first introduced to the mouse. Top, PETH of individual trials; bottom, average ($n = 80$ trials). **c**, Average PETH of sessions when punishment was first introduced ($n = 8$ mice). Inset, average BFPVN response to the first air puff presentation ($n = 8$). **d**, Schematic diagram of foot shock presentation. **e**, PETH aligned to foot shocks in an example session ($n = 10$ trials). **f**, Average PETH of foot shock response ($n = 4$ animals; 10 sessions each). **g**, Schematic diagram of fox odor presentation. **h**, PETH aligned to fox odor presentation in an example session. **i**, Average PETH of fox odor response in PVBFNs. ($n = 4$ animals; 1 session each; note the longer time scale). Errorshades represent SEM.

4. Mapping of the input-output of BFPVNs has been performed in previous work. I wonder about the necessity of repeating these experiments.

We thank the Reviewer for this question, which prompted us to better explain how our tracing studies relate to previous work. Monosynaptic inputs to BFPVNs have been mapped by the Yang Dan lab (Do et al., 2016). However, that study focused on comparing BF cell types and did not further specify the location of the injections. Since previous studies suggested a topographic organization of the BF (Saper, 1984; Zaborszky et al., 2012) with a specific input-output logic (Zaborszky et al., 2013; Gielow and Zaborszky, 2017), we decided to reveal the monosynaptic inputs specifically to the PV neurons of the horizontal diagonal band of Broca (HDB), a part of the BF that may be implicated in cognitive control (Lin and Nicolelis, 2008; Kim et al., 2015; Yang et al., 2017). Indeed, the input patterns of HDB BFPVNs only partially overlapped with those found for a broader population by Do et al., most importantly, lateral hypothalamic inputs dominated over striatal inputs.

Outputs of the BFPVNs were also mapped in the same study by Do et al., however, we again found strong differences when focused on HDB BFPVNs. In an elegant anatomical study by Agostinelli et al., they mapped the cell types and nucleus-specific outputs of BF neurons in subcortical areas (Agostinelli et al., 2019). However, that study lumped the GABAergic cell types by using Vgat as a marker. Still, HDB GABAergic neurons in Agostinelli et al. showed almost perfect concordance with HDB BFPVNs in our study when focusing on subcortical areas in contrast to the marked difference from the Do et al. study, again confirming that it is important to take the BF topography into account. We are not aware of a comprehensive mapping of the projections of HDB BFPVNs or HDB GABAergic neurons in cortical areas, but they were suggested to project to the medial prefrontal cortex. While we confirmed the presence of these projections, to our surprise, a denser projection was revealed targeting the retrosplenial cortex.

In sum, we are unaware of a comprehensive input-output mapping of HDB BFPVNs, which revealed broad targeting of limbic areas (medial septum, retrosplenial cortex, hippocampus, supramammillary nucleus) while receiving strong inputs from centers known to process aversive information (lateral hypothalamus, lateral septum, medial septum, median raphe nucleus). We expanded the discussion on previous BF GABAergic tracing/mapping studies.

5. The authors used fiber photometry recording to measure the calcium signal from axons of BFPVNs, and found a broadcasting pattern. Further manipulation experiments should be performed to identify the downstream brain regions that mediate the behavior effect.

We thank the Reviewer for this important question, aiming at revealing potential differences in the downstream impact of BFPVNs in distinct target areas. We demonstrated that BFPVN axons in the dorsal CA1, retrosplenial cortex and medial septum exhibit similar characteristics in their average responses to punishment. Closer examination of individual traces of single trials and sessions (added to the manuscript as Fig.7b-i) may suggest that these punishment responses, while similar on average, were more consistent and higher in magnitude in the CA1 compared to the other two targets.

We tested whether punishment responses in the target areas predicted performance in the next session and found a positive correlation with next-day lick rate difference of mice in the CA1

(Figure R4) but not in the MS and retrosplenial cortex. This also suggests that BFPVNs projecting to the CA1 may be especially important in mediating the learning effect we observed. Interestingly, we did not find a significant correlation at the somatic level, which might be due to recording a mixed population of BFPVNs projecting to different targets.

Figure R4. The magnitude of BFPVN punishment response in CA1 predicted behavioral performance (indexed by anticipatory lick rate difference) in the next training session.

Following up on the next point of Reviewer #1, we tested downstream impact of optogenetic stimulation of BFPVNs in the CA1, MS and retrosplenial cortex *in vitro* (see also our response to point #6). We found strong inhibitory synaptic inputs on PV+ hippocampal neurons (Figure R5), known to robustly control hippocampal function and associated behavior (Fuchs et al., 2007). These results again suggest a prominent impact of the CA1-projecting BFPVN population on downstream neurons.

Finally, we appreciate the Reviewer's suggestion of addressing this question by performing projection-specific optogenetic manipulations. However, while the evidence above points to CA1 as a potentially important target, our fiber photometry recordings are also consistent with a

distributed circuit that mediates the behavioral effect, limiting the strength of the hypothesis that inhibiting the axons of HDB BFPVNs in one of the downstream areas would reproduce the effect of somatic inhibition. Additionally, technical limitations prevent a 100% axonal blockade (incomplete viral infection, questionable efficiency of presently available optogenetic actuators in axonal inhibition) and retrograde strategies are limited due to the large size of the innervated structures. We would like to also note that training on probabilistic Pavlovian conditioning takes about two weeks, where animals have to be trained in a head-fixed setup, limiting the number of animals that can be trained in parallel. Given these limitations, we had to conclude that this experiment is beyond the possibilities of this study and the timeframe of this revision.

6. The authors used EM to identify synaptic contact of BFPVNs. However, these experiments did not provide too much information. They should perform a more systematic analysis of the postsynaptic target neurons. I think EM may not be the right method to use for this purpose, and an alternative is to perform brain slice electrophysiology recording.

We thank the Reviewer for this suggestion. We took the Reviewer's advice and performed channelrhodopsin-assisted circuit mapping experiments in the dorsal CA1, medial septum and retrosplenial cortex (Figure R5). These experiments confirmed all connections revealed by EM. Furthermore, we found that BFPVN inhibitory inputs to the RSC showed smaller amplitude compared to the MS and CA1 as well as longer latency to peak and slower rise time (Figure R5g). Importantly, when we examined the short-term synaptic plasticity of BFPVN inputs, we found that inputs to the RSC neurons were characterized by short-term synaptic facilitation, while MS and CA1 neurons showed strong short-term synaptic depression in a frequency dependent manner.

These results suggest that BFPVN output to its prominent target regions might undergo reconfiguration depending on their firing rate, which varies with behavioral context. We included these new experiments in the manuscript (Fig.8 and corresponding text).

Figure R5. Functional synaptic contacts of BFPVNs show target area specific short-term plasticity. **a**, To enable channelrhodopsin-assisted circuit mapping experiments, PV-Cre mice ($n = 7$) were injected with Cre-dependent AAV, to selectively express ChR2 in BFPVN axonal projections. **b**, After sufficient expression time of 3-6 weeks, acute brain slices were prepared to perform slice electrophysiology recording from the retrosplenial cortex (RSC), CA1 area of the dorsal hippocampus and the medial septum (MS). All recordings were performed in the presence of fast excitatory synaptic transmission blockers AP-5 and NBQX (50 μ M and 20 μ M, respectively) at -65 mV, using intracellular solution containing high chloride (see Methods). **c**, Representative example of postsynaptic inhibitory currents recorded from a neuron in response to the optogenetic stimulation of BFPVN fibers (average of 20 stimulation periods overlaid in black). The inhibitory response was eliminated in the presence of the GABA-A receptor blocker gabazine (10 μ M). **d-f**, Representative confocal images of responsive cells from the retrosplenial cortex, medial septum, and the hippocampus, respectively (recorded cells are in green, BFPVN axonal projections are in red). Using post-hoc immunohistochemistry, functional contacts on PV+ cells in RSC, CA1 and MS, CR+ in the hippocampus and ChAT+ cells in the MS (not shown) were identified in line with our EM data. **g**, Top left, optogenetically evoked IPSCs were

significantly smaller on RSC neurons ($n = 5$) compared to MS neurons ($n = 6$) and showed a tendentious difference compared to CA1 neurons ($n = 7$) as well (median amplitude values in pA for MS, RSC and CA1, respectively: 108.04, 32.34, 98.0; RSC vs. MS, $p = 0.017$; RSC vs. CA1, $p = 0.148$). Latency to peak was higher in the RSC compared to the other two regions (median values in ms for MS, RSC and CA1, respectively: 2.9, 4.3, 3.3; RSC vs. MS, $p = 0.051$; RSC vs. CA1, $p = 0.030$). In line with these, 10-90% rise-time was shortest in the CA1 (median values in ms for MS, RSC and CA1, respectively: 1.29, 1.6, 0.6); RSC vs. CA1, $p = 0.010$). We found no differences in 90-10% decay-time (median values in ms for MS, RSC and CA1, respectively: 10.35, 13.0, 8.4). All statistical comparisons were performed using Mann-Whitney U-test. **h**, To uncover the short-term synaptic dynamics of BFPVN fibers terminating in the examined target regions, we delivered two consecutive light pulses at different frequencies to calculate paired pulse ratios (PPR). Responses from representative hippocampal and retrosplenial neurons are shown (black, average of 20 stimulation periods). Note the prominent short-term synaptic depression of the BFPVN input onto the hippocampal cell (black arrow), and the short-term facilitation up to 20 Hz of the BFPVN input onto the retrosplenial cell (blue arrow). **i**, Summary data of PPR at different frequencies on retrosplenial (blue), hippocampal (black) and medial septal neurons (red). Median PPR values @ 5, 10, 20 and 40 Hz, MS: 0.64, 0.74, 0.82, 0.82; RSC: 0.95, 1.19, 1.66, 0.26; CA1: 0.70, 0.67, 0.50, 0.20; RSC vs. CA1 @ 10 Hz, $p = 0.008$, RSC vs. MS @ 10 Hz, $p = 0.030$, RSC vs. CA1 @ 20 Hz, $p = 0.051$, Mann-Whitney U-test.

7. For the fiber photometry recording experiments, the raw signal trace should be provided rather than only showing an averaged plot.

We added single trial and single session fiber photometry data to Fig. 7 (see below).

Figure R6. Single trial and single session calcium measurements of BFPVN cell bodies and axonal projections. a, Example session recorded in the HDB. Top, single-trial dff traces; bottom, average PETH aligned to air puff punishments. **b,** All single sessions of HDB recordings (n = 19), sorted according to descending amplitudes. **c,** Example session recorded in the MS. Top, single-trial dff traces; bottom, average PETH aligned to air puff punishments. **d,** All single sessions (n = 8) of MS recordings, sorted according to descending amplitudes. **e,** Example session recorded in the CA1. Top, single-trial dff traces; bottom, average PETH aligned to air puff punishments. **f,** All single sessions of CA1 recordings (n = 9), sorted according to descending amplitudes. **g,** Example session recorded in the RSC. Top, single-trial dff traces; bottom, average PETH aligned to air puff punishments. **h,** All single sessions of RSC recordings (n = 5), sorted according to descending amplitudes. Errorshades indicate SEM.

8. The author found that BFPVNs receive inputs from aversion-coding nuclei. How do the authors know these neurons actually provide the aversive information to BFPVNs?

Activation of HDB BFPVNs after aversive stimuli might come from one specific input, or multiple inputs. Since we identified multiple nuclei that are known to process such stimuli and innervate HDB BFPVNs through excitatory connections, we hypothesized that more than one input contributes to the aversion-coding of BFPVNs. Nevertheless, we cannot rule out the possibility that one of these inputs dominates over the others, and we have limited knowledge on whether other HDB neurons participate in this process through local connections onto BFPVNs. Of note, lateral hypothalamic inputs dominated over striatal inputs to HDB BFPVNs, while these two sources were balanced in a more general BF PVN population (Do et al., 2016), which may underlie the robust punishment responses in comparison to reward-related activity. We now discuss these limitations in the Discussion section.

9. Further details of the optogenetic inhibition experiments should be provided. At least, a daily learning curve from the two groups during learning should be provided

We added learning curves and behavioral comparison in earlier stages of learning as Figure S7a-b.

Figure R7. Differential learning in Control and ArchT-inhibited mice. Left, learning curve of Control and ArchT animals (anticipatory lick rate difference plotted as a function of training days). Middle and right panels, PETH and bar graph showing no anticipatory lick rate difference between

Control and ArchT mice at an earlier training stage (before introducing punishment; Wilcoxon signed-rank test; n.s., $p > 0.05$).

10. how do the authors explain that BFPVNs broadcast aversive information, but activating them did not cause aversive behavior?

Information about aversive outcomes has multiple relevance for the animals, likely involving partly overlapping, but partly divergent circuits. First, aversive outcomes typically (but not mandatorily, see (Yawata et al., 2023)) evoke avoidance behavior, engaging effector circuits and eventually muscles. A number of nuclei and cell types have been shown to be involved in active avoidance (Faget et al., 2018; Wenzel et al., 2018; Szőnyi et al., 2019; Stephenson-Jones et al., 2020; Manning et al., 2021). Therefore, we first tested whether BFPVNs also contribute to these functions, but we found that this was not the case. Second, animals have to learn from aversive information, e.g. form stimulus-outcome associations that will then be stored in memory. Since mice report the recall of such associations by differential anticipatory behavior in Pavlovian conditioning, we could test whether HDB BFPVNs are necessary to form such associations, where we found that this was indeed the case. Third, aversive information also leads to arousal, attention and increased vigilance, to which HDB BFPVNs might also contribute (Yang et al., 2017). Since they were also hypothesized to contribute to arousal on a relatively fast timescale (Kim et al., 2015; Yang et al., 2017), we put forth the hypothesis that HDB BFPVNs might specifically increase ‘attention for learning’, thus mediating associative learning processes through increasing cortical excitability at specific target regions. Thus, it is likely that these different aspects of aversive stimuli (avoidance, arousal, attention, learning) are mediated by at least partially separable brain circuits. Please see also our response to the related point #2 of Reviewer #2. We expanded our Discussion to include these thoughts.

Specific comments:

In Fig. 3, the cartoon in panel A provides little information. Also, a diagram to indicate the timing of laser stimulation should be added.

We thank the Reviewer for the suggestion. We included a timeline for the laser stimulation to make the panel more informative.

Fiber photometry recording of calcium signal is not ‘imaging’. This should be corrected.

We changed the wording.

In some experiments, the number of mice equals 2?

In one experiment, the monosynaptic inputs from the median raphe region were identified by cell type from two mice undergoing mono-transsynaptic rabies tracing. Since we did not expect a large variability in input proportions (confirmed by the consistency across the two mice investigated)

and we did not draw conclusions from the proportional distribution of median raphe inputs, we consider this still informative despite the low sample size.

Responses to comments from Reviewer #2

Hegedüs et al investigated how basal forebrain parvalbumin neurons (BFPVNs) play a role in associative learning. Previous studies suggest that BFPVNs regulate cortical gamma oscillation and promote wakefulness. However, how these neurons participate in awake behavior remains unclear. To answer this question, the authors first recorded the activity of phototagged BFPVNs in HDB while head-fixed mice performed an auditory probabilistic Pavlovian conditioning task, in which two auditory cues were associated with different outcome probabilities (80% reward / 10% airpuff / 10% omission for cue1 and 25% reward / 65% airpuff / 10% omission for cue2). BFPVNs showed phasic activation to airpuff, and such responses were not modulated by expectation. Optogenetic inhibition of BFPVNs during punishment impairs behavioral discrimination of the two cues. However, optogenetic activation of BFPVNs did not generate place aversion. The authors further mapped the inputs and outputs of these neurons, and show that BFPVNs receive inputs from aversion-related nuclei, such as medial raphe. GCamp6 photometry recording confirmed that BFPVNs were excited by airpuff, and broadcast this signal to its main downstream targets, including medial septum, hippocampus and retrosplenial cortex. Based on these results, the authors concluded that BFPVNs broadcast aversive information to downstream targets to promote arousal and mediate learning from negative experience. While phototagging of BFPVNs in behaving animals provides valuable information about how these neurons operate under behavioral contexts, there are considerable inconsistencies in the results that made me question their interpretation.

We thank the Reviewer for their thorough and constructive evaluation of our study, based on which we performed novel experiments and analyses to strengthen the conclusions drawn from our experiments.

Major comments:

1. A key result of the study is that optogenetic inhibition of BFPVNs during airpuff impairs behavioral discrimination of the two cues. This manipulation eliminated differences in anticipatory licking patterns and reaction time (Figure 3). However, even if ArchT inhibition during airpuff presentation abolished the effect of the negative outcome, the two cues were still different in terms reward probabilities (80% vs 25%). Shouldn't the differential reward alone be sufficient to endow different behavioral response patterns to the two cues? The lack of behavioral difference in Fig 3C cannot be explained by the lack of punishment effect alone. Instead, it would be more compatible with a failure to discriminate between the two cues (sounds).

We thank the Reviewer for raising this important point. Our general experience is that rodents learn faster if both reward and punishment are involved as compared to using reward alone, confirming previous studies (Wächter et al., 2009; Steel et al., 2016; Chen et al., 2018; Yin et al., 2023). However, we agree with the Reviewer, and also found in other tasks, that reward alone,

even if slower, is typically sufficient to induce learning. Therefore, to test the effect of BFPVN inhibition during punishment presentation, we controlled training time and mice underwent a two-week-long standardized training protocol. Within these two weeks, control mice developed discriminative anticipatory licking behavior, unlike the inhibited group. However, it is possible, or even expected, that both groups would have learned the contingencies during longer training.

We would like to stress that PV neurons were only inhibited during air puffs, 0.2-0.4 seconds after the termination of the one-second-long tone presentation, and several seconds ($> 2.5s$) before the next tone presentation, making it unlikely that the inhibition interfered with sensory processing. To make this clearer, we included the timeline of optogenetic inhibition in Fig. 4a, as also suggested by Reviewer #1.

To further corroborate this result, we directly compared lick rates of inhibited and control groups for Cue 1 and Cue 2. We found no significant difference for licking after Cue 1 (likely reward cue, $p = 0.373$, Mann-Whitney U-test), whereas ArchT-expressing mice licked significantly more after Cue 2 (likely punishment cue, $p = 0.013$) compared to Control mice (Figure R8a). Therefore, the difference between Control and ArchT mice stems from a lack of suppression of anticipatory activity when punishment can be expected, reflecting incorrect expectations specifically related to the punishment-predicting stimulus. In contrast, impaired stimulus discrimination would lead to an intermediate expectation after the different cues, expected to alter lick rate to Cue 1, too.

We also tested whether ArchT mice were less active, or reacted slower, which would be expected if detecting of the sounds were impaired. We did not find significant differences in reaction time to the likely reward tone between ArchT and Control mice ($p = 0.715$), suggesting that ArchT mice were not generally less active or anticipating reward less. On the other hand, reaction time to Cue2 was shorter in ArchT mice ($p = 0.0128$) likely reflecting incorrect expectations related exclusively to the punishment-predicting stimulus (Figure R8b).

Figure R8. Direct comparison of lick rates and reaction times after the two cues in Control and ArchT-inhibited mice. **a**, Lick rate during Cue1 and Cue2 presentation in Control and ArchT mice. **b**, Reaction time to Cue1 and Cue2 in Control and ArchT mice. *, $p < 0.05$, two sided Mann-Whitney U-test.

We added these plots to Fig.S7.

2. The lack of effect on real-time place preference/aversion (Fig S4) further support that BFPVNs neither encode aversive nor rewarding information. Are they any example of aversive-encoding neurons whose activation does not lead to behavioral aversion?

Information about aversive outcomes has multiple relevance for the animals, likely involving partly overlapping, but partly divergent circuits. First, aversive outcomes typically (but not mandatorily, see (Yawata et al., 2023)) evoke avoidance behavior, engaging effector circuits and eventually muscles. A number of nuclei and cell types (mostly glutamatergic) have been shown to be involved in active avoidance (Faget et al., 2018; Szőnyi et al., 2019; Stephenson-Jones et al., 2020; Manning et al., 2021; Barbano et al., 2023). Second, animals have to learn from aversive information, e.g. form stimulus-outcome associations that will then be stored in memory (Manning et al., 2021). Third, aversive information also leads to arousal, attention and increased vigilance.

These are different processes induced by aversive sensory information; therefore, it is expected that they be mediated by at least partially separable circuits. For example, basal forebrain cholinergic neurons respond rapidly and reliably to aversive stimuli including shocks and air puffs, but their stimulation evokes neither avoidance nor approach (Hangya et al., 2015; McKenna et al., 2021). Indeed, they are thought to contribute to the learning aspects of outcomes by controlling cortical plasticity (Kilgard and Merzenich, 1998; Froemke et al., 2007; Seol et al., 2007; Gu and Yakel, 2011; Hangya et al., 2015). Being part of the same anatomical structure, BFPVNs are likely also involved in cognitive processing rather than direct motor effector functions. In line with this, they were hypothesized to contribute to arousal on both slower and faster timescales (Kim et al., 2015; Xu et al., 2015; Yang et al., 2017; McKenna et al., 2020). The lack of place avoidance after photostimulation of BFPVNs combined with impaired learning caused by their photoinhibition confirms this hypothesis. We propose that HDB BFPVNs might specifically increase ‘attention for learning’, thus mediating associative learning processes through increasing cortical excitability at specific target areas, probably by disinhibition. We expanded our Discussion to include these thoughts.

3. Is it possible that BFPVNs were responding to sound intensity instead of airpuff? Airpuff delivery should induce a much louder sound than either cue1 or cue2. Such auditory responses were reported in basal forebrain cholinergic neurons and might not be unexpected to observe in BFPVNs (Guo W, Robert B, Polley DB. The Cholinergic Basal Forebrain Links Auditory Stimuli with Delayed Reinforcement to Support Learning. Neuron. 2019. doi:10.1016/j.neuron.2019.06.024).

We thank the Reviewer for raising this question. To address this, we tested whether HDB BFPVNs respond to aversive stimuli in other sensory modalities: foot shock and fox odor. We found robust BFPVN calcium responses to these stimuli by fiber photometry, presented in the figure below, which we included in the manuscript as Fig. 3.

Figure R9 (same as Figure R3). BFPVNs respond to aversive stimuli of different modality (air puff, foot shock, fox odor) in naïve mice. **a**, Schematic diagram of introducing punishment during the Pavlovian conditioning task. **b**, Example of a session when punishment was first introduced to the mouse. Top, PETH of individual trials; bottom, average ($n = 80$ trials). **c**, Average PETH of sessions when punishment was first introduced ($n = 8$ mice). Inset, average BFPVN response to the first air puff presentation ($n = 8$). **d**, Schematic diagram of foot shock presentation. **e**, PETH aligned to foot shocks in an example session ($n = 10$ trials). **f**, Average PETH of foot shock response ($n = 4$ animals; 10 sessions each). **g**, Schematic diagram of fox odor presentation. **h**, PETH aligned to fox odor presentation in an example session. **i**, Average PETH of fox odor response in PVBFNs. ($n = 4$ animals; 1 session each; note the longer time scale). Errorshades indicate SEM.

With respect to sound responses of different BF cell types, indeed both cholinergic and GABAergic neurons respond to auditory stimuli in the basal forebrain; however, most of their responses to auditory stimuli are learned responses that strongly depend on stimulus salience, confirmed by multiple studies, including the cited influential paper from the Polley lab (Lin and Nicolelis, 2008;

Hangya et al., 2015; Guo et al., 2019; Robert et al., 2021; Hegedüs et al., 2023). We found that HDB BFPVNs, like BFCNs, also ‘encode sound and aversive stimuli’ (from the ‘Highlights’ of Guo et al.: ‘Optotagged cholinergic basal forebrain neurons encode sound and aversive stimuli’), but with substantially different proportions and temporal dynamics: their responses are dominated by a strong response to air puff as compared to the reward or outcome-predicting auditory cues, and they have a 77 ± 10 ms (mean \pm standard error) response latency to air puff compared to the very fast (18 ms) cholinergic activation.

Finally, intensive sounds evoke a startle response from mice and are generally considered aversive. Thus, we agree with Reviewer that the loud sound of air puffs may contribute to the activation of BFPVNs. Nevertheless, HDB BFPVN responses to shock and fox odor indicate that BFPVN responses to aversive stimuli are multimodal, and not solely driven by the auditory modality.

Could BFPVN manipulations exert their effect through modulating BF cholinergic neurons?

This theoretical possibility is unlikely based on previous literature, which demonstrated that unlike rich local connectivity of other basal forebrain cell types (glutamatergic and cholinergic neurons), BFPVNs do not innervate BFCNs or BFSOMs, and provide only weak and sparse connections to local glutamatergic neurons (Xu et al., 2015; Yang et al., 2017). In line with this, Kim et al. found that the BFPVNs impact cortical gamma oscillations independent of BFCNs (Kim et al., 2015). We added this note to the Discussion section.

4. Several previous studies have emphasized the role of BFPVNs in arousal through their projections to the cerebral cortex (Kim et al, PNAS, 2015; Xu et al, Nat Neurosci 2015; McKenna et al 2020). But the current results only find frontal cortex to be a minor target of BFPVNs (line 274), and there is little effect that BFPVNs would lead to increased arousal. In Fig S4 (RTPP), there was also no effect on movement pattern. How do the authors interpret this discrepancy?

We thank the Reviewer for this insightful question. Indeed, McKenna, Brown, McCarley, Basheer and others have a line of elegant publications linking BFPVNs to arousal. They revealed that this arousal could occur at short timescales (‘microarousals’), likely controlling cortical excitability according to attentional needs (Kim et al., 2015; Yang et al., 2017; McKenna et al., 2020). This is in line with some of the earliest recording studies of the BF, in which Richardson and DeLong proposed that the BF might mediate cortical arousal at different time scales (Richardson and DeLong, 1991), confirmed by Buzsaki et al. (Buzsaki et al., 1988). In contrast, BFPVNs have not been linked to the type of ‘general arousal’ associated with noradrenergic and cholinergic systems (Aston-Jones and Waterhouse, 2016; Thiele and Bellgrove, 2018; Maness et al., 2022). Our results are consistent with this view.

Nevertheless, one might indeed expect stronger mPFC projections based on previous literature. We would like to note that our tracings did reveal projections to medial orbital cortex and to the infralimbic cortex; however, we found denser projections in the CA1 and retrosplenial cortex, thus we focused more on the latter areas. Some differences might be due to anatomical topography (Zaborszky et al., 2012; Agostinelli et al., 2019): we used injection sites slightly anterior compared

to Kim et al. (Kim et al., 2015) closer to the VDB/HDB border (Fig. 6a) compared to posterior HDB/SI injections in the cited paper to avoid infecting VP PV neurons, which is a mixed population of GABAergic and glutamatergic neurons with different connectivity and likely different function (Knowland et al., 2017; Faget et al., 2018).

We now added *in vitro* recordings performing channelrhodopsin-assisted circuit mapping of BFPVN projections and found that BFPVN impact is particularly strong in the CA1 and MS when compared to the RSC (Figure R5). Furthermore, we found, that BFPVN impact on target regions might be strongly shaped by differential short-term synaptic plasticity: frequency-dependent short-term facilitation on neocortical, and short-term synaptic depression on hippocampal and medial septal neurons. Thus, the impact of BFPVN activity across target areas could dynamically change with varying behavioral and brain states.

Additionally, behavioral performance indexed by anticipatory lick rate difference after cue presentation on the following day correlated positively with the magnitude of punishment-associated BFPVN calcium responses in the CA1 measured by fiber photometry (Figure R4). These data further strengthen the hypothesis that BFPVNs are important in learning about aversive outcomes, given the known roles of the hippocampus in episodic memory processes. Again, this is reconcilable with the view that BFPVNs lead to disinhibition of pyramidal neurons through innervating mostly interneurons, which could be a circuit-level substrate of (local) arousal or attention.

We added these thoughts to the Discussion section.

5. PV neurons are ubiquitously present in all brain circuits. The location of BFPVNs should be carefully characterized in its various experiments (phototagging, optogenetics, GCamp6, viral tracing) to make sure that the results point to the same population of PVNs. The use of 300nl GCamp6 virus raises the concerns of a large infection volume beyond the HDB region. Do the author expect some of the results may result from heterogeneity of the underlying neuronal population?

We absolutely agree with the Reviewer that in the case of such a widespread cell type as PV, one should be very careful with viral injections. Therefore, we used 20-50 nl injection volumes for tracing experiments, 70-100 nl for optogenetic excitation and inhibition experiments, and 100 nl injection volumes for GCaMP experiments (erroneously stating 300 nl in the previous version of the manuscript, for which we apologize). We indeed injected large volumes for optotagging likely resulting in broad infection; this was a conscious choice to increase the yield of optogenetic tagging, where spatial specificity was achieved by precise tetrode targeting.

Also, we carefully checked all injections, and added a figure showing injection sites of the fiber photometry experiments to the revised manuscript (Fig. R10, added to Fig.S8 of the manuscript; see also Fig.1f, Fig.5a, Fig.6a and Fig.7a, all corresponding to the same anterior HDB area).

Figure R10. Injection sites in the HDB for fiber photometry recordings of HDB BFPVN projections. **a**, Representative fluoromicrograph of a GCaMP6s injection in a PV-Cre mouse. Scale bar, 500 μm . **b**, reconstruction of all injection sites in the HDB.

Lastly, we would like to point out that the GCaMP experiments resulted in fiber photometry traces, including somatic recordings, that were concordant with electrophysiology recordings of optogenetically identified HDB BFPVNs and expressed little heterogeneity. To further visualize this, we added individual trial and session data for fiber photometry, as also suggested by Reviewer #1 (Figure R6, added to Fig. 7).

Other comments:

1. *What's distinction between no-lick period and foreperiod in the task design (Fig 1a)? They seem redundant.*

We thank the Reviewer for this comment. We merged these two states and thus simplified the diagram.

2. *The difference in licking pattern at the population level (Fig 1d) was much smaller than the example session in Fig 1c. The authors might want to consider showing a more representative session in Fig 1c.*

We replaced the example session.

3. *Figure 1f and its inset do not match up. Which area was magnified?*

The inset shows a different section that shows the electrolytic lesion better (whereas the entire track is less visible). We clarified this in the figure legend.

4. Phototagging provides highly valuable basic physiological characterizations of PV neurons in the BF. The authors should provide a table showing the raw firing rate characteristics of all phototagged neurons (baseline firing rates, ISI distribution, peak firing rate....), in addition to the normalized firing rate in Fig 1j.

We agree with the Reviewer regarding the value of these data; therefore, we provided a table (Table R2, included in the manuscript as Table S4) with the baseline firing rate and punishment-evoked peak firing rate and latency of all identified BFPVNs. Furthermore, we uploaded the ISI histogram and autocorrelogram of all BFPVNs as a supplementary file.

Cellid	Baseline firing rate (Hz)	Peak firing rate after punishment (Hz)	Peak latency (ms)
'HDB17_170720a_4.2'	13.16	29.38	119
'HDB17_170723a_7.1'	11.14	22.89	20
'HDB17_170724a_7.2'	10.10	27.45	85
'HDB17_170725a_5.2'	21.93	47.88	91
'HDB17_170805a_5.1'	20.47	60.61	105
'HDB17_170807a_5.2'	18.24	34.08	158
'HDB17_170810a_3.1'	30.95	48.78	89
'HDB17_170810a_5.1'	26.10	89.49	83
'HDB17_170811a_3.2'	26.15	43.48	74
'HDB17_170812a_4.1'	12.48	21.06	73
'HDB17_170904a_4.2'	20.14	104.48	87
'HDB17_170904a_6.2'	2.78	33.08	19
'HDB17_170906a_6.3'	1.62	11.35	70
HDB17_170912a_6.1'	7.69	13.29	21
'HDB17_170928a_2.1'	4.36	36.47	86
'HDB17_170928a_4.1'	17.67	99.99	90
'HDB17_170928a_4.2'	8.05	14.62	16
'HDB17_171010a_2.1'	4.13	53.05	81
'HDB23_180221a_3.2'	28.38	39.83	15
'HDB23_180223a_3.1'	14.71	22.74	12
'HDB23_180223a_5.3'	4.26	22.80	66
'HDB34_190113a_7.1'	18.18	30.28	182
'HDB34_190115a_5.1'	16.53	40.53	15
'HDB34_190117a_4.1'	2.91	5.46	33
'HDB34_190118a_4.1'	4.25	8.39	97
'HDB34_190207a_8.1'	14.87	28.86	197
'HDB30_181002a_2.2'	2.85	8.52	95
HDB17_170811a_4.2'	11.62		

'HDB17_170812a_3.1'	23.68		
'HDB17_170912a_4.1'	35.32		
'HDB23_180225a_3.1'	16.24		
'HDB34_190115a_4.2'	4.73		
'HDB34_190123a_2.1'	53.69		
'HDB34_190127a_2.1'	18.92		
'HDB34_190207a_6.1'	17.97		
'HDB30_181002a_2.1'	5.26		
Average ± standard error	15.32 ± 1.84	36.99 ± 5.24	77.0 ± 9.52

Table R2. Baseline firing rate for all identified BFPVNs (n = 36) and punishment-evoked peak firing rate of punishment-activated BFPVNs (n = 27) along with peak latency of punishment response.

5. *If BFPVN responses to airpuff were not modulated by surprise (Fig S2), what could explain the difference in response amplitude and baseline firing rates between the first and second half of the task (Fig 2g-i)?*

We thank the Reviewer for this question. One relatively straightforward explanation could be that the aversive quality of the air puffs decreases with repeated exposure, leading to decreased responses and consequentially probably decreased learning rate controlled by BFPVNs. However, multiple variables change gradually with session progression, including thirst, motivation, fatigue and stimulus novelty, which could potentially all contribute to the observed slow changes. We added a discussion on this topic in the Discussion section of the manuscript.

6. *The differences in anticipatory licking in Fig 3b seemed to arise immediately after cue onset in the control group. Yet it only reached statistical significance much later (0.6-1.1s), and the RTs (Fig 3d) were also much slower in both groups compared to the population average in Fig 3b, 3c. This discrepancy should be addressed.*

We looked into this in detail and found that reaction times show a rather large variability, consistent with the literature. Short reaction times are considerably different from the median, leading to early onset times on averaged time-resolved graphs (some level of temporal smoothing during PETH calculation adds to this effect – added to Methods). However, partly due to the same variability, standard deviations are slightly larger for earlier time points of the post-stimulus window, which, while not visually obvious in Fig.2b, lowers statistical significance when running the permutation tests on the ROC curves. We added a note on this to the figure legend (Fig.S7).

7. *The detailed of the optogenetic inhibition should be provided: was inhibition delivered throughout all stages of the training? Or just the last stage? Was the behavioral difference present during the first session of the last stage? Were there any behavioral difference at earlier stages of training?*

Optogenetic inhibition was delivered during air puff presentations from the first introduction of air puffs; we clarified this in the Methods. We also added a timeline to the optogenetic inhibition experiment (Fig.4a). The behavior differences developed in the last stage of training after the introduction of the air puffs (Figure R7, added to Figure S7 of the manuscript).

8. *Vglut3 immunostaining in Fig 4f: How does Vglut3 staining help determine whether the neuron expresses Vglut3? I thought Vglut3 should be present mostly in the synapse and not in the soma, just like Vglut2?*

Unlike many vesicular transporters, Vglut3 shows considerable somatic expression as well (Nickerson Poulin et al., 2006).

9. *The spread of GCamp6 viral expression should be further characterized to minimize the effect of injecting PV neurons in nearby brain regions.*

We characterized the injection sites in Fig.S8 (see also response to major point #5).

10. *Fig 5d: vGlut1+ neuron image missing.*

Thank you for spotting this discrepancy. The figure legend contained an error, which has been corrected.

11. *GCamp6 responses sometimes start before time zero (Fig 6b, S6). Where the traces smoothed? By what kind of smoothing kernel?*

We used a Gaussian kernel (width = 100 ms) to smooth the PETHs. We added this information to the figure legend and the Methods.

References

- Agostinelli LJ, Geerling JC, Scammell TE (2019) Basal forebrain subcortical projections. *Brain Struct Funct* 224:1097–1117.
- Aston-Jones G, Waterhouse B (2016) Locus coeruleus: From global projection system to adaptive regulation of behavior. *Brain Res* 1645:75–78.
- Avila I, Lin S-C (2014) Distinct neuronal populations in the basal forebrain encode motivational salience and movement. *Front Behav Neurosci* 8:421.
- Bañuelos C, Kittleson JR, LaNasa KH, Galiano CS, Roth SM, Perez EJ, Long JM, Roberts MT, Fong S, Rapp PR (2023) Cognitive Aging and the Primate Basal Forebrain Revisited: Disproportionate GABAergic Vulnerability Revealed. *J Neurosci* 43:8425–8441.
- Barbano MF, Zhang S, Chen E, Espinoza O, Mohammad U, Alvarez-Bagnarol Y, Liu B, Morales M (2023) Lateral Hypothalamic Glutamatergic Inputs to VTA Glutamatergic

Neurons Mediate Prioritization of Innate Defensive Behavior over Feeding. *IBRO Neurosci Reports* 15:S935–S936.

- Bland BH, Oddie SD, Colom L V (1999) Mechanisms of Neural Synchrony in the Septohippocampal Pathways Underlying Hippocampal Theta Generation. *J Neurosci* 19:3223–3237.
- Buzsaki G, Bickford R, Ponomareff G, Thal L, Mandel R, Gage F (1988) Nucleus basalis and thalamic control of neocortical activity in the freely moving rat. *J Neurosci* 8:4007–4026.
- Chaves-Coira I, García-Magro N, Zegarra-Valdivia J, Torres-Alemán I, Núñez Á (2023) Cognitive Deficits in Aging Related to Changes in Basal Forebrain Neuronal Activity. *Cells* 12:1477.
- Chen X, Holland P, Galea JM (2018) The effects of reward and punishment on motor skill learning. *Curr Opin Behav Sci* 20:83–88.
- Cohen JY, Haesler S, Vong L, Lowell BB, Uchida N (2012) Neuron-type-specific signals for reward and punishment in the ventral tegmental area. *Nature* 482:85–88.
- Crouse RB, Kim K, Batchelor HM, Girardi EM, Kamaletdinova R, Chan J, Rajebhosale P, Pittenger ST, Role LW, Talmage DA, Jing M, Li Y, Gao X-B, Mineur YS, Picciotto MR (2020) Acetylcholine is released in the basolateral amygdala in response to predictors of reward and enhances the learning of cue-reward contingency. *Elife* 9:1–31.
- Do JP, Xu M, Lee S, Chang W, Zhang S, Chung S, Yung TJ, Fan JL, Miyamichi K, Luo L, Dan Y (2016) Cell type-specific long-range connections of basal forebrain circuit. *Elife* 5:1–17.
- Dwyer TA, Servatius RJ, Pang KCH (2007) Noncholinergic Lesions of the Medial Septum Impair Sequential Learning of Different Spatial Locations. *J Neurosci* 27:299–303.
- Etter G, van der Veldt S, Manseau F, Zarrinkoub I, Trillaud-Doppia E, Williams S (2019) Optogenetic gamma stimulation rescues memory impairments in an Alzheimer's disease mouse model. *Nat Commun* 10:1–11.
- Faget L, Zell V, Souter E, McPherson A, Ressler R, Gutierrez-Reed N, Yoo JH, Dulcis D, Hnasko TS (2018) Opponent control of behavioral reinforcement by inhibitory and excitatory projections from the ventral pallidum. *Nat Commun* 9:849.
- Freund TF, Antal M (1988) GABA-containing neurons in the septum control inhibitory interneurons in the hippocampus. *Nature* 336:403–405.
- Froemke RC, Merzenich MM, Schreiner CE (2007) A synaptic memory trace for cortical receptive field plasticity. *Nature* 450:425–429.
- Fuchs EC, Zivkovic AR, Cunningham MO, Middleton S, LeBeau FEN, Bannerman DM, Rozov A, Whittington MA, Traub RD, Rawlins JNP, Monyer H (2007) Recruitment of Parvalbumin-Positive Interneurons Determines Hippocampal Function and Associated Behavior. *Neuron* 53:591–604.
- Gielow MR, Zaborszky L (2017) The Input-Output Relationship of the Cholinergic Basal Forebrain. *Cell Rep* 18:1817–1830.

- Gritti I, Henny P, Galloni F, Mainville L, Mariotti M, Jones BE (2006) Stereological estimates of the basal forebrain cell population in the rat, including neurons containing choline acetyltransferase, glutamic acid decarboxylase or phosphate-activated glutaminase and colocalizing vesicular glutamate transporters. *Neuroscience* 143:1051–1064.
- Gu Z, Yakel JL (2011) Timing-dependent septal cholinergic induction of dynamic hippocampal synaptic plasticity. *Neuron* 71:155–165.
- Guo W, Robert B, Polley DB (2019) The Cholinergic Basal Forebrain Links Auditory Stimuli with Delayed Reinforcement to Support Learning. *Neuron* 103:1164-1177.e6.
- Hangya B, Borhegyi Z, Szilagyi N, Freund TF, Varga V (2009) GABAergic Neurons of the Medial Septum Lead the Hippocampal Network during Theta Activity. *J Neurosci* 29:8094–8102.
- Hangya B, Ranade SP, Lorenc M, Kepecs A (2015) Central Cholinergic Neurons Are Rapidly Recruited by Reinforcement Feedback. *Cell* 162:1155–1168.
- Hegedüs P, Sviatkó K, Király B, Martínez-Bellver S, Hangya B (2023) Cholinergic activity reflects reward expectations and predicts behavioral responses. *iScience* 26:105814.
- Iaccarino HF, Singer AC, Martorell AJ, Rudenko A, Gao F, Gillingham TZ, Mathys H, Seo J, Kritskiy O, Abdurrob F, Adaikkan C, Canter RG, Rueda R, Brown EN, Boyden ES, Tsai L-H (2016) Gamma frequency entrainment attenuates amyloid load and modifies microglia. *Nature* 540:230–235.
- Joshi A, Salib M, Viney TJ, Dupret D, Somogyi P (2017) Behavior-Dependent Activity and Synaptic Organization of Septo-hippocampal GABAergic Neurons Selectively Targeting the Hippocampal CA3 Area. *Neuron* 96:1342-1357.e5.
- Kilgard MP, Merzenich MM (1998) Cortical Map Reorganization Enabled by Nucleus Basalis Activity. *Science* (80-) 279:1714–1718.
- Kim T, Thankachan S, McKenna JT, McNally JM, Yang C, Choi JH, Chen L, Kocsis B, Deisseroth K, Strecker RE, Basheer R, Brown RE, McCarley RW (2015) Cortically projecting basal forebrain parvalbumin neurons regulate cortical gamma band oscillations. *Proc Natl Acad Sci* 112:3535–3540.
- Knowland D, Lilascharoen V, Pacia CP, Shin S, Wang EHJ, Lim BK (2017) Distinct Ventral Pallidal Neural Populations Mediate Separate Symptoms of Depression. *Cell* 170:284-297.e18.
- Laszlovszky T, Schlingloff D, Hegedüs P, Freund TF, Gulyás A, Kepecs A, Hangya B (2020) Distinct synchronization, cortical coupling and behavioral function of two basal forebrain cholinergic neuron types. *Nat Neurosci* 23:992–1003.
- Lin S-C, Nicolelis M a L (2008) Neuronal ensemble bursting in the basal forebrain encodes salience irrespective of valence. *Neuron* 59:138–149.
- Lozano-Montes L, Dimanico M, Mazloun R, Li W, Nair J, Kintscher M, Schneggenburger R, Harvey M, Rainer G (2020) Optogenetic Stimulation of Basal Forebrain Parvalbumin Neurons Activates the Default Mode Network and Associated Behaviors. *Cell Rep*

33:108359.

- Maness EB, Burk JA, McKenna JT, Schiffino FL, Strecker RE, McCoy JG (2022) Role of the locus coeruleus and basal forebrain in arousal and attention. *Brain Res Bull* 188:47–58.
- Manning EE, Bradfield LA, Iordanova MD (2021) Adaptive behaviour under conflict: Deconstructing extinction, reversal, and active avoidance learning. *Neurosci Biobehav Rev* 120:526–536.
- Martorell AJ, Paulson AL, Suk H-J, Abdurrob F, Drummond GT, Guan W, Young JZ, Kim DN-W, Kritskiy O, Barker SJ, Mangena V, Prince SM, Brown EN, Chung K, Boyden ES, Singer AC, Tsai L-H (2019) Multi-sensory Gamma Stimulation Ameliorates Alzheimer’s-Associated Pathology and Improves Cognition. *Cell* 177:256-271.e22.
- McGeer PL, McGeer EG, Singh VK, Chase WH (1974) Choline acetyltransferase localization in the central nervous system by immunohistochemistry. *Brain Res* 81:373–379.
- McKenna JT, Thankachan S, Uygun DS, Shukla C, McNally JM, Schiffino FL, Cordeira J, Katsuki F, Zant JC, Gamble MC, Deisseroth K, McCarley RW, Brown RE, Strecker RE, Basheer R (2020) Basal Forebrain Parvalbumin Neurons Mediate Arousals from Sleep Induced by Hypercarbia or Auditory Stimuli. *Curr Biol* 30:2379-2385.e4.
- McKenna JT, Yang C, Bellio T, Anderson-Chernishof MB, Gamble MC, Hulverson A, McCoy JG, Winston S, Hodges E, Katsuki F, McNally JM, Basheer R, Brown RE (2021) Characterization of basal forebrain glutamate neurons suggests a role in control of arousal and avoidance behavior. *Brain Struct Funct* 226:1755–1778.
- McKenna JT, Yang C, Franciosi S, Winston S, Abar KK, Rigby MS, Yanagawa Y, McCarley RW, Brown RE (2013) Distribution and intrinsic membrane properties of basal forebrain GABAergic and parvalbumin neurons in the mouse. *J Comp Neurol* 521:1225–1250.
- Nickerson Poulin A, Guerci A, El Mestikawy S, Semba K (2006) Vesicular glutamate transporter 3 immunoreactivity is present in cholinergic basal forebrain neurons projecting to the basolateral amygdala in rat. *J Comp Neurol* 498:690–711.
- Richardson RT, DeLong MR (1991) Electrophysiological Studies of the Functions of the Nucleus Basalis in Primates. In: *Advances in experimental medicine and biology*, pp 233–252.
- Robert B, Kimchi EY, Watanabe Y, Chakoma T, Jing M, Li Y, Polley DB (2021) A functional topography within the cholinergic basal forebrain for encoding sensory cues and behavioral reinforcement outcomes. *Elife* 10:1–28.
- Roland JJ, Janke KL, Servatius RJ, Pang KCH (2014) GABAergic neurons in the medial septum-diagonal band of Broca (MSDB) are important for acquisition of the classically conditioned eyeblink response. *Brain Struct Funct* 219:1231–1237.
- Saper CB (1984) Organization of cerebral cortical afferent systems in the rat. II. Magnocellular basal nucleus. *J Comp Neurol* 222:313–342.
- Seol GH, Ziburkus J, Huang S, Song L, Kim IT, Takamiya K, Hugarir RL, Lee HK, Kirkwood A (2007) Neuromodulators control the polarity of spike-timing-dependent synaptic

- plasticity. *Neuron* 55:919–929.
- Serafin M, Williams S, Khateb A, Fort P, Mühlethaler M (1996) Rhythmic firing of medial septum non-cholinergic neurons. *Neuroscience* 75:671–675.
- Stanley EM, Fadel JR, Mott DD (2012) Interneuron loss reduces dendritic inhibition and GABA release in hippocampus of aged rats. *Neurobiol Aging* 33:431.e1-431.e13.
- Steel A, Silson EH, Stagg CJ, Baker CI (2016) The impact of reward and punishment on skill learning depends on task demands. *Sci Rep* 6:36056.
- Stephenson-Jones M, Bravo-Rivera C, Ahrens S, Furlan A, Xiao X, Fernandes-Henriques C, Li B (2020) Opposing Contributions of GABAergic and Glutamatergic Ventral Pallidal Neurons to Motivational Behaviors. *Neuron* 105:921-933.e5.
- Szónyi A, Zichó K, Barth AM, Gönczi RT, Schlingloff D, Török B, Sipos E, Major A, Bardóczi Z, Sos KE, Gulyás AI, Varga V, Zelena D, Freund TF, Nyiri G (2019) Median raphe controls acquisition of negative experience in the mouse. *Science* 366:eaay8746.
- Thiele A, Bellgrove MA (2018) Neuromodulation of Attention. *Neuron* 97:769–785.
- Wächter T, Lungu O V., Liu T, Willingham DT, Ashe J (2009) Differential Effect of Reward and Punishment on Procedural Learning. *J Neurosci* 29:436–443.
- Wenzel JM, Oleson EB, Gove WN, Cole AB, Gyawali U, Dantrassy HM, Bluett RJ, Dryanovski DI, Stuber GD, Deisseroth K, Mathur BN, Patel S, Lupica CR, Cheer JF (2018) Phasic Dopamine Signals in the Nucleus Accumbens that Cause Active Avoidance Require Endocannabinoid Mobilization in the Midbrain. *Curr Biol* 28:1392-1404.e5.
- Xu M, Chung S, Zhang S, Zhong P, Ma C, Chang WC, Weissbourd B, Sakai N, Luo L, Nishino S, Dan Y (2015) Basal forebrain circuit for sleep-wake control. *Nat Neurosci* 18:1641–1647.
- Yang C, Thankachan S, McCarley RW, Brown RE (2017) The menagerie of the basal forebrain: how many (neural) species are there, what do they look like, how do they behave and who talks to whom? *Curr Opin Neurobiol* 44:159–166.
- Yawata Y, Shikano Y, Ogasawara J, Makino K, Kashima T, Ihara K, Yoshimoto A, Morikawa S, Yagishita S, Tanaka KF, Ikegaya Y (2023) Mesolimbic dopamine release precedes actively sought aversive stimuli in mice. *Nat Commun* 14:2433.
- Yin C, Gao T, Li B (2023) The effect of combining punishment and reward can transfer to opposite motor learning. *Fogt N, ed. PLoS One* 18:e0282028.
- Yoder RM, Pang KCH (2005) Involvement of GABAergic and cholinergic medial septal neurons in hippocampal theta rhythm. *Hippocampus* 15:381–392.
- Zaborszky L, Csordas A, Mosca K, Kim J, Gielow MR, Vadasz C, Nadasdy Z (2013) Neurons in the Basal Forebrain Project to the Cortex in a Complex Topographic Organization that Reflects Corticocortical Connectivity Patterns: An Experimental Study Based on Retrograde Tracing and 3D Reconstruction. *Cereb Cortex*.
- Zaborszky L, van den Pol A, Gyengesi E (2012) The Basal Forebrain Cholinergic Projection

System in Mice. In: *The Mouse Nervous System*, 1st ed. (Watson C, Paxinos G, Puelles L, eds), pp 684–718. Amsterdam: Elsevier.

Reviewers' Comments:

Reviewer #1:

Remarks to the Author:

Thanks the authors for addressing my concerns. The current manuscript is much improved. I have no more further concerns.

Reviewer #2:

Remarks to the Author:

The authors have done an excellent job in this revision. The additional experiments and analyses provide a coherent and comprehensive picture supporting that BFPVN broadcast aversive information to multiple downstream regions to facilitate learning about aversive outcomes. The detailed experiments and careful characterizations elegantly demonstrated the function of PV neurons located specifically in the HDB region. The final results are very convincing.

My only minor comment is that, when discussing the potential role of BFPVNs in serving 'attention for learning', this idea must be restricted to conveying aversive information only. Appetitive information should be conveyed by other neural circuits.

Rebuttal letter: Parvalbumin-expressing basal forebrain neurons mediate learning from negative experience

Structure:

Reviewer comments: *black, italic*

Our responses: blue

Reviewer #1:

Thanks the authors for addressing my concerns. The current manuscript is much improved. I have no more further concerns.

We thank the Reviewer for the constructive suggestions and the positive evaluation of our work.

Reviewer #2:

The authors have done an excellent job in this revision. The additional experiments and analyses provide a coherent and comprehensive picture supporting that BFPVN broadcast aversive information to multiple downstream regions to facilitate learning about aversive outcomes. The detailed experiments and careful characterizations elegantly demonstrated the function of PV neurons located specifically in the HDB region. The final results are very convincing.

We greatly appreciate the Reviewer's positive evaluation of our work and the important comments, which have significantly contributed to the manuscript.

My only minor comment is that, when discussing the potential role of BFPVNs in serving 'attention for learning', this idea must be restricted to conveying aversive information only. Appetitive information should be conveyed by other neural circuits.

We agreed with the Reviewer's suggestion and thus modified the corresponding sentence accordingly:

'We propose that HDB BFPVNs might specifically increase attention for aversive learning, thus mediating associative learning processes through increasing cortical excitability at specific target areas, probably by disinhibition.'

Reviewer #2 (Remarks on code availability):

The code is very detailed and contains all aspects of analyses in this study.

We thank the Reviewer for appreciating our work.